# LEARNING TO REASON OVER VISUAL OBJECTS

**Shanka Subhra Mondal***
Princeton University
Princeton, NJ
smondal@princeton.edu

**Taylor W. Webb***
University of California, Los Angeles
Los Angeles, CA
taylor.w.webb@gmail.com

**Jonathan D. Cohen**
Princeton University
Princeton, NJ
jdc@princeton.edu

**\* Equal Contribution**

## ABSTRACT

A core component of human intelligence is the ability to identify abstract patterns inherent in complex, high-dimensional perceptual data, as exemplified by visual reasoning tasks such as Raven's Progressive Matrices (RPM). Motivated by the goal of designing AI systems with this capacity, recent work has focused on evaluating whether neural networks can learn to solve RPM-like problems. Previous work has generally found that strong performance on these problems requires the incorporation of inductive biases that are specific to the RPM problem format, raising the question of whether such models might be more broadly useful. Here, we investigated the extent to which a general-purpose mechanism for processing visual scenes in terms of objects might help promote abstract visual reasoning. We found that a simple model, consisting only of an object-centric encoder and a transformer reasoning module, achieved state-of-the-art results on both of two challenging RPM-like benchmarks (PGM and I-RAVEN), as well as a novel benchmark with greater visual complexity (CLEVR-Matrices). These results suggest that an inductive bias for object-centric processing may be a key component of abstract visual reasoning, obviating the need for problem-specific inductive biases.

## 1 INTRODUCTION

Human reasoning is driven by a capacity to extract simple, low-dimensional abstractions from complex, high-dimensional inputs. We perceive the world around us in terms of objects, relations, and higher order patterns, allowing us to generalize beyond the sensory details of our experiences, and make powerful inferences about novel situations Spearman (1923); Gick & Holyoak (1983); Lake et al. (2017). This capacity for abstraction is particularly well captured by visual analogy problems, in which the reasoner must abstract over the superficial details of visual inputs, in order to identify a common higher order pattern (Gentner, 1983; Holyoak, 2012). A particularly challenging example of these kinds of problems are the Raven's Progressive Matrices (RPM) problem sets (Raven, 1938), which have been found to be especially diagnostic of human reasoning abilities (Snow et al., 1984).

A growing body of recent work has aimed to build learning algorithms that capture this capacity for abstract visual reasoning. Much of this previous work has revolved around two recently developed benchmarks – the Procedurally Generated Matrices (PGM) (Barrett et al., 2018), and the RAVEN dataset (Zhang et al., 2019a) – consisting of a large number of automatically generated RPM-like problems. As in RPM, each problem consists of a $3 \times 3$ matrix populated with geometric forms, in which the bottom right cell is blank. The challenge is to infer the abstract pattern that governs the relationship along the first two columns and/or rows of the matrix, and use that inferred pattern to 'fill in the blank', by selecting from a set of choices. As can be seen in Figure 1, these problems can be quite complex, with potentially many objects per cell, and multiple rules per problem, yielding a highly challenging visual reasoning task.

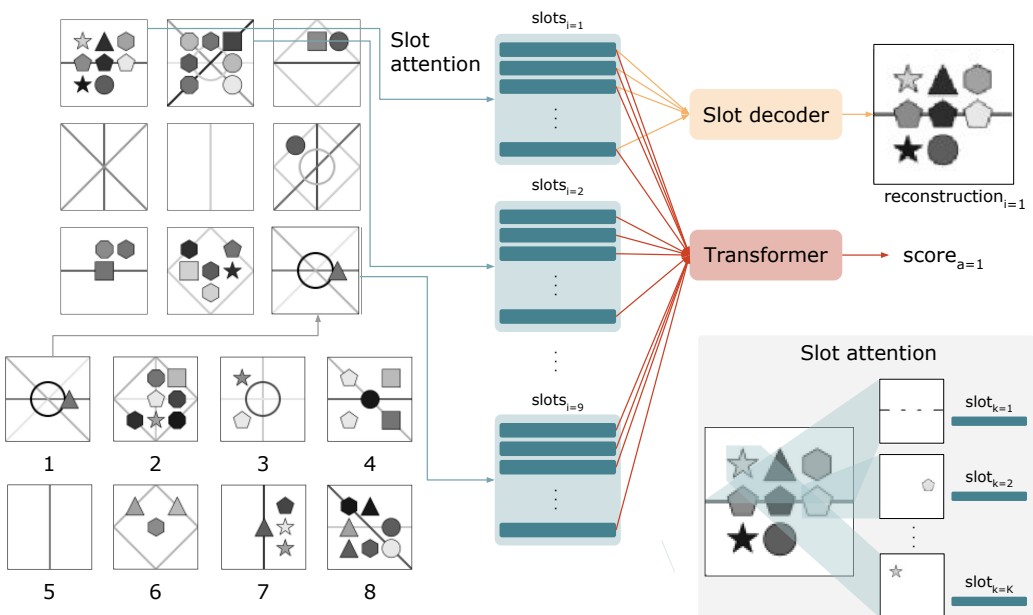

Figure 1: Slot Transformer Scoring Network (STSN). STSN combines slot attention, an object-centric encoding method, and a transformer reasoning module. Slot attention decomposes each image panel into a set of $K$ slots, which are randomly initialized and iteratively updated through competitive attention over the image. STSN assigns a score to each of the 8 potential answers, by independently evaluating the combination of each answer choice together with the 8 context panels. For each answer choice, slots are extracted from that choice, and the context panels, and these slots are concatenated to form a sequence that is passed to the transformer, which then generates a score. The scores for all answer choices are passed through a softmax in order to compute the task loss $\mathcal{L}_{task}$. Additionally, the slots for each image panel are passed through a slot decoder, yielding a reconstruction of that image panel, from which the reconstruction loss $\mathcal{L}_{recon}$ is computed.

There is substantial evidence that human visual reasoning is fundamentally organized around the decomposition of visual scenes into objects (Duncan, 1984; Pylyshyn, 1989; Peters & Kriegeskorte, 2021). Objects offer a simple, yet powerful, low-dimensional abstraction that captures the inherent compositionality underlying visual scenes. Despite the centrality of objects in visual reasoning, previous works have so far not explored the use of object-centric representations in abstract visual reasoning tasks such as RAVEN and PGM, or at best have employed an imprecise approximation to object representations based on spatial location.

Recently, a number of methods have been proposed for the extraction of precise object-centric representations directly from pixel-level inputs, without the need for veridical segmentation data (Greff et al., 2019; Burgess et al., 2019; Locatello et al., 2020; Engelcke et al., 2021). While these methods have been shown to improve performance in some visual reasoning tasks, including question answering from video (Ding et al., 2021) and prediction of physical interactions from video Wu et al. (2022), previous work has not addressed whether this approach is useful in the domain of *abstract* visual reasoning (i.e., visual analogy). To address this, we developed a model that combines an object-centric encoding method, *slot attention* (Locatello et al., 2020), with a generic transformer-based reasoning module (Vaswani et al., 2017). The combined system, termed the *Slot Transformer Scoring Network* (STSN, Figure 1) achieves state-of-the-art performance on both PGM and I-RAVEN (a more challenging variant of RAVEN), despite its general-purpose architecture, and lack of task-specific augmentations. Furthermore, we developed a novel benchmark, the *CLEVR-Matrices* (Figure 2), using a similar RPM-like problem structure, but with greater visual complexity, and found that STSN also achieves state-of-the-art performance on this task. These results suggest that object-centric encoding is an essential component for achieving strong abstract visual reasoning, and indeed may be even more important than some task-specific inductive biases.

## 2 RELATED WORK

Since the introduction of the PGM (Barrett et al., 2018) and RAVEN (Zhang et al., 2019a) datasets, a number of methods have been proposed for learning to solve RPM-like problems Barrett et al. (2018); Steenbrugge et al. (2018); Van Steenkiste et al. (2019); Zhang et al. (2019b); Zheng et al. (2019); Spratley et al. (2020); Jahrens & Martinetz (2020); Wang et al. (2020); Wu et al. (2020); Benny et al. (2021); Hu et al. (2021); Zhuo & Kankanhalli (2022). Though significant progress has been made, the best performing methods generally rely on inductive biases that are specifically tailored to the RPM problem format. For instance, the Scattering Compositional Learner (SCL) (Wu et al., 2020), arguably the best current model (achieving strong performance on both PGM and I-RAVEN), assumes that rules are independently applied in each feature dimension, with no interaction between features. Similarly, the Multi-Scale Relation Network (MRNet) (Benny et al., 2021), which achieves strong performance on PGM, explicitly builds the row-wise and column-wise structure of RPM problems into its architecture. These approaches raise the question of whether problem-specific inductive biases are necessary to achieve strong performance on these problems.

Here, we explore the utility of a more general-purpose inductive bias – a mechanism for processing visual scenes in terms of objects. In contrast, most previous approaches to solving RPM-like problems have operated over embeddings of entire image panels, and thus likely fail to capture the compositional structure of such multi-object visual inputs. Some work has attempted to approximate object-centric representations, for instance by treating spatial location as a proxy for objects (Wu et al., 2020), or by employing encodings at different spatial scales (Benny et al., 2021) (therefore preferentially capturing larger vs. smaller objects), but it is not clear that these approximations extract precise object-centric representations, especially in problems with many overlapping objects, such as PGM.

Recently, a number of methods have been proposed to address the challenging task of annotation-free object segmentation (Greff et al., 2019; Burgess et al., 2019; Locatello et al., 2020; Engelcke et al., 2021). In this approach, the decomposition of a visual scene into objects is treated as a latent variable to be inferred in the service of a downstream objective, such as autoencoding, without access to any explicit segmentation data. Here, we used the slot attention method (Locatello et al., 2020), but our approach should be compatible with other object-centric encoding methods.

Our method employs a generic transformer (Vaswani et al., 2017) to perform reasoning over the object-centric representations extracted by slot attention. This approach allows the natural permutation invariance of objects to be preserved in the reasoning process. A few other recent efforts have employed systems that provide object-centric representations as the input to a transformer network (Ding et al., 2021; Wu et al., 2022), most notably ALOE (Attention over Learned Object Embeddings (Ding et al., 2021)), which used a different object encoding method (MONet (Burgess et al., 2019)). Such systems have exhibited strong visual reasoning performance in some tasks, such as question answering from video, that require processing of relational information. Here, we go beyond this work, to test: a) the extent to which object-centric processing can subserve more *abstract* visual reasoning, involving the processing of higher-order relations, as required for visual analogy tasks such as PGM and I-RAVEN; and b) whether this approach obviates the need for problem-specific inductive biases that have previously been proposed for these tasks.

## 3 APPROACH

### 3.1 PROBLEM DEFINITION

Each RPM problem consists of a $3 \times 3$ matrix of panels in which each panel is an image consisting of varying numbers of objects with attributes like size, shape, and color. The figures in each row or column obey a common set of abstract rules. The last panel (in the third row and column), is missing and must be filled from a set of eight candidate panels so as to best complete the matrix according to the abstract rules. Formally, each RPM problem consists of 16 image panels $X = \{x_i\}_{i=1}^{16}$, in which the first 8 image panels are context images $X_c = \{x_i\}_{i=1}^{8}$ (i.e., all panels in the $3 \times 3$ problem matrix except the final blank panel), and the last 8 image panels are candidate answer images $X_a = \{x_i\}_{i=9}^{16}$. The task is to select $y$, the index for the correct answer image.

## 3.2 Object-centric Encoder

STSN employs slot attention (Locatello et al., 2020) to extract object-centric representations. Slot attention first performs some initial processing of the images using a convolutional encoder, producing a feature map, which is flattened to produce $\mathbf{inputs} \in \mathbb{R}^{N \times D_{inputs}}$, where $N = H \times W$ (the height and width of the feature map), and $D_{inputs}$ is the number of channels. Then, the slots $\mathbf{slots} \in \mathbb{R}^{K \times D_{slot}}$ are initialized, to form a set of $K$ slot embeddings, each with dimensionality $D_{slot}$. We set the value of $K$ to be equal to the maximum number of objects possible in a given image panel (based on the particular dataset). For each image, the slots are randomly initialized from a distribution $\mathcal{N}(\mu, \mathrm{diag}(\sigma)) \in \mathbb{R}^{K \times D_{slot}}$ with shared mean $\mu \in \mathbb{R}^{D_{slot}}$ and variance $\sigma \in \mathbb{R}^{D_{slot}}$ (each of which are learned). The slots are then iteratively updated based on a transformer-style attention operation. Specifically, each slot emits a query $\mathrm{q}(\mathbf{slots}) \in \mathbb{R}^{K \times D_{slot}}$ through a linear projection, and each location in the feature map emits a key $\mathrm{k}(\mathbf{inputs}) \in \mathbb{R}^{N \times D_{slot}}$ and value $\mathrm{v}(\mathbf{inputs}) \in \mathbb{R}^{N \times D_{slot}}$. A dot product query-key attention operation followed by softmax is then used to generate the attention weights $\mathbf{attn} = \mathrm{softmax}(\frac{1}{\sqrt{D_{slot}}} \mathrm{k}(\mathbf{inputs}) \cdot \mathrm{q}(\mathbf{slots})^{\top})$, and a weighted mean of the values $\mathbf{updates} = \mathbf{attn} \cdot \mathrm{v}(\mathbf{inputs})$ is used to update the slot representations using a Gated Recurrent Unit (Cho et al., 2014), followed by a residual MLP with ReLU activations. More details can be found in Locatello et al. (2020). After $T$ iterations of slot attention, the resulting slots are passed through a reasoning module, that we describe in the following section.

In order to encourage the model to make use of slot attention in an object-centric manner, we also included a slot decoder to generate reconstructions of the original input images. To generate reconstructions, we first used a spatial broadcast decoder (Watters et al., 2019) to generate both a reconstructed image $\tilde{x}_k$, and a mask $m_k$, for each slot. We then generated a combined reconstruction, by normalizing the masks across slots using a softmax, and using the normalized masks to compute a weighted average of the slot-specific reconstructions.

## 3.3 Reasoning Module

After object representations are extracted by slot attention, they are then passed to a transformer (Vaswani et al., 2017). For each candidate answer choice $x_a \in \{x_i\}_{i=9}^{16}$, the transformer operates over the slots obtained from the 8 context images $\mathbf{slots}_{x_{1..8}}$, and the image for that answer choice $\mathbf{slots}_{x_a}$. We flattened the slots over the dimensions representing the number of slots and images, such that, for each candidate answer, the transformer operated over $\mathrm{flatten}(\mathbf{slots}_{x_{1..8}}, \mathbf{slots}_{x_a}) \in \mathbb{R}^{9K \times D_{slot}}$. We then applied Temporal Context Normalization (TCN) (Webb et al., 2020), which has been shown to significantly improve out-of-distribution generalization in relational tasks, over the flattened sequence of slots. To give the model knowledge about which slot representation corresponded to which row and column of the matrix, we added a learnable linear projection $\mathbb{R}^6 \to \mathbb{R}^{D_{slot}}$ from one-hot encodings of the row and column indices (after applying TCN). We concatenated a learned CLS token (analogous to CLS token in Devlin et al. (2018)) of dimension $D_{slot}$, before passing it through a transformer with $L$ layers and $H$ self-attention heads. The transformed value of the CLS token was passed through a linear output unit to generate a score for each candidate answer image, and the scores for all answers were passed through a softmax to generate a prediction $\hat{y}$.

## 3.4 Optimization

The entire model was trained end-to-end to optimize two objectives. First, we computed a reconstruction loss $\mathcal{L}_{recon}$, the mean squared error between the 16 image panels and their reconstructed outputs. Second, we computed a task loss $\mathcal{L}_{task}$, the cross entropy loss between the target answer index and the softmax-normalized scores for each of the candidate answers. These two losses were combined to form the final loss $\mathcal{L} = \lambda * \mathcal{L}_{recon} + \mathcal{L}_{task}$, where $\lambda$ is a hyperparameter that controls the relative strength of the reconstruction loss.

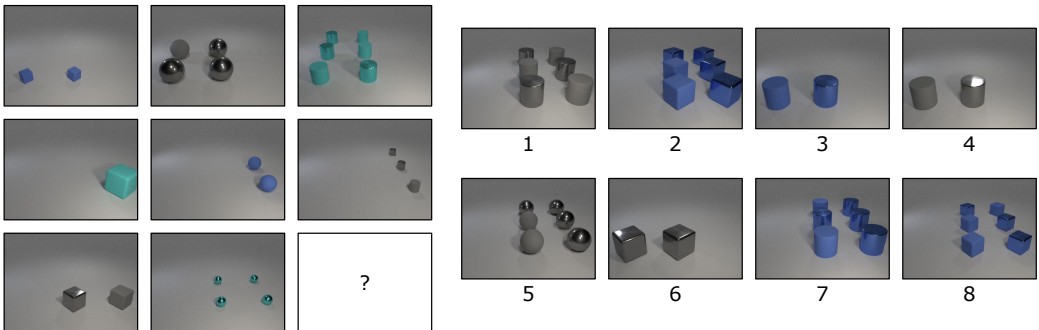

Figure 2: Example problem from our proposed CLEVR-Matrices dataset. Problems are governed by RPM-like problem structure, but with greater visual complexity (rendered using approach similar to CLEVR dataset (Johnson et al., 2017)). This particular problem is an example of the 'Location' problem type. The reader is encouraged to identify the correct answer, and rule for each attribute.

## 4 EXPERIMENTS

### 4.1 DATASETS

**PGM.** The PGM dataset was introduced by Barrett et al. (2018), and consists of problems belonging to eight different regimes with different generalization difficulty. Each matrix problem in PGM is defined by the abstract structure $\mathcal{S} = \{[r, o, a] : r \in \mathcal{R}, o \in \mathcal{O}, a \in \mathcal{A}\}$, where $\mathcal{R} = \{$progression, XOR, AND, OR, consistent union$\}$ are the set of rules (note that 'consistent union' is also referred to as 'distribution-of-3'), $\mathcal{O} = \{$shape, line$\}$ are the set of objects, and $\mathcal{A} = \{$size, type, position, color, number$\}$ are the set of attributes. Each regime consists of 1.2M training problems, 20K validation problems, and 200K testing problems. Due to the enormous size of the dataset we focused on the neutral, interpolation, and extrapolation regimes. In the neutral regime, the training and test sets are sampled from the same underlying distribution, whereas the interpolation and extrapolation regimes both involve out-of-distribution generalization. Given the set of feature values for each attribute, the interpolation regime involves training on all even-indexed feature values and testing on all odd-indexed values, and the extrapolation regime involves training on the lower half of feature values and testing on the upper half of feature values. More details can be found in Barrett et al. (2018).

**I-RAVEN.** The RAVEN dataset was introduced by Zhang et al. (2019a), with problems belonging to seven different configurations. These configurations are defined by the spatial layout of the elements in each panel, ranging from low visual complexity (e.g., the 'Center' configuration, in which each panel contains just a single object in the center of the image), to high visual complexity (e.g., the 'O-IG' configuration, in which each panel contains an outer object surrounding an inner grid of objects). Some configurations have multiple components $\mathcal{C}$ to which separate rules can be bound. Thus, each problem in RAVEN is defined by the abstract structure $\mathcal{S} = \{[r, c, a] : r \in \mathcal{R}, c \in \mathcal{C}, a \in \mathcal{A}\}$, where $\mathcal{R} = \{$constant, progression, arithmetic, distribution-of-3$\}$ are the set of rules, $\mathcal{C}$ are the set of components (depending on the particular configuration), and $\mathcal{A} = \{$number, position, size, type, color$\}$ are the set of attributes. There are a total of 42K training problems, 14K validation problems, and 14K testing problems. We trained STSN jointly on all configurations in RAVEN.

It was subsequently discovered that the original RAVEN dataset employed a biased method for generating candidate answers, that could be exploited so as to achieve near perfect performance by only viewing these candidate answers (i.e., ignoring the problem itself) (Hu et al., 2021). To address this, Hu et al. (2021) proposed the Impartial RAVEN (I-RAVEN) dataset, with an unbiased procedure for generating candidate answers. As with most recent work in this domain, we performed our evaluation on I-RAVEN.

**CLEVR-Matrices.** We created a novel dataset of RPM-like problems using realistically rendered 3D shapes, based on source code from CLEVR (a popular visual-question-answering dataset) (Johnson et al., 2017). Problems were formed from objects of three shapes (cube, sphere, and cylinder), three sizes (small, medium, and large), and eight colors (gray, red, blue, green, brown, purple, cyan,

and yellow). Objects were placed on a $3 \times 3$ grid of locations (such that there was a maximum of 9 objects in each panel), which was oriented randomly in each problem. Lighting was varied randomly between each panel, and objects were randomly assigned one of two textures (metal or rubber). Rules were independently sampled for shape, color, and size, from the set $\mathcal{R} = \{$null, constant, distribution-of-3$\}$. Location was determined based on three different problem types. In the first problem type ('Logic'), locations were determined based on a logical rule sampled from $\mathcal{R} = \{$AND, OR, XOR$\}$. In the second problem type ('Location'), locations were determined based on a rule sampled from $\mathcal{R} = \{$constant, distribution-of-3, progression$\}$. In the third problem type ('Count'), the count of objects in each panel was determined based on a rule sampled from $\mathcal{R} = \{$constant, distribution-of-3, progression$\}$, and locations were randomly sampled to instantiate that count. Example problems are shown in Figure 2 and Section A.5. Answer choices were generated using the attribute bisection tree algorithm proposed by Hu et al. (2021), which was used to generate answer choices for I-RAVEN. Our dataset thus does not contain the biases identified in the original RAVEN dataset. We generated 20K problems for each type, including 16K for training, 2K for validation, and 2K for testing. We trained STSN jointly on all three problems types.

## 4.2 BASELINES

We compared our model to several baselines, as detailed in Tables 1- 3. To the best of our knowledge, these baselines include the current best performing models on the I-RAVEN and PGM benchmarks. We didn't use any auxiliary information (i.e., training to explicitly label the underlying rules), and hence for fair comparison we only compared to baselines that didn't use auxiliary loss.

There are too many baselines to describe them each in detail, but here we briefly describe the best performing baselines. The baseline that achieved the best overall performance was the Scattering Compositional Learner (SCL) (Wu et al., 2020). SCL employs an approximate form of object segmentation based on fixed spatial locations in a convolutional feature map, followed by a dual parameter-sharing scheme, in which a shared MLP (shared across 'objects') is used to generate object embeddings, and another shared MLP (shared across attributes) is used to classify rules for each attribute. We also compare against the Multi-Layer Relation Network (MLRN) (Jahrens & Martinetz, 2020) and the Multi-scale Relation Network (MRNet) (Benny et al., 2021), both of which achieved strong results on PGM. MLRN builds on the Relation Network (Santoro et al., 2017), which uses a shared MLP to compute learned relation vectors for all pairwise comparisons of a set (in this case, the set of embeddings for all image panels in a problem). MLRN passes the output of one RN to another RN, thus allowing second-order relations to be modeled. MRNet creates image embeddings at different spatial scales, allowing it to approximate segmentation of larger vs. smaller objects, and then computes both row-wise and column-wise rule embeddings, which are aggregated across both rows/columns and spatial scales.

## 4.3 EXPERIMENTAL DETAILS

We give a detailed characterization of all hyperparameters and training details for our models in Section A.2. We employed both online image augmentations (random rotations, flips, and brightness changes) and dropout (in the transformer), when training on I-RAVEN (details in Section A.2). We also trained both SCL and MLRN on CLEVR-Matrices, and compared to two alternative versions of SCL on I-RAVEN, one that employed the same image augmentations, TCN, and dropout employed by our model, and another that combined SCL with slot attention (also with image augmentations, TCN and dropout) referred to as 'Slot-SCL'

For I-RAVEN, to be consistent with previous work (Wu et al., 2020), we report results from the best out of 5 trained models. Similarly, for CLEVR-Matrices, we report results from the best out of 3 trained models for STSN, SCL, and MLRN. For PGM, we only trained 1 model on the neutral regime, 1 model on the interpolation regime, and 1 model on the extrapolation regime, due to the computational cost of training models on such a large dataset.

For the PGM neutral regime, we pretrained the convolutional encoder, slot attention, and slot decoder on the reconstruction objective with the neutral training set, and fine-tuned while training on the primary task. For the PGM interpolation regime, all model components were trained end-to-end from scratch. For the the PGM extrapolation regime, we employed a simultaneous dual-training scheme, in which the convolutional encoder, slot attention, and slot decoder were trained on recon-

Table 1: Results on I-RAVEN.

| Model | Test Accuracy (%) | | | | | | | |
|---|---|---|---|---|---|---|---|---|
| | Average | Center | 2Grid | 3Grid | L-R | U-D | O-IC | O-IG |
| LSTM (Hu et al., 2021) | 18.9 | 26.2 | 16.7 | 15.1 | 14.6 | 16.5 | 21.9 | 21.1 |
| WReN (Hu et al., 2021) | 23.8 | 29.4 | 26.8 | 23.5 | 21.9 | 21.4 | 22.5 | 21.5 |
| MLRN (Jahrens & Martinetz, 2020) | 29.8 | 38.8 | 32.0 | 27.8 | 23.5 | 23.4 | 32.9 | 30.0 |
| LEN (Zheng et al., 2019) | 39.0 | 45.5 | 27.9 | 26.6 | 44.2 | 43.6 | 50.5 | 34.9 |
| ResNet (Hu et al., 2021) | 40.3 | 44.7 | 29.3 | 27.9 | 51.2 | 47.4 | 46.2 | 35.8 |
| Wild ResNet (Hu et al., 2021) | 44.3 | 50.9 | 33.1 | 30.8 | 53.1 | 52.6 | 50.9 | 38.7 |
| CoPINet (Zhang et al., 2019b) | 46.3 | 54.4 | 33.4 | 30.1 | 56.8 | 55.6 | 54.3 | 39.0 |
| SRAN (Hu et al., 2021) | 63.9 | 80.1 | 53.3 | 46.0 | 72.8 | 74.5 | 71.0 | 49.6 |
| Slot-SCL | 90.4 | 98.8 | 94.1 | 80.3 | 92.9 | 94.0 | 94.9 | 78.0 |
| SCL (Wu et al., 2020) | 95.0 | **99.0** | 96.2 | 89.5 | 97.9 | 97.1 | 97.6 | 87.7 |
| SCL + dropout, augmentations, TCN | 95.5 | 98.2 | **96.4** | **90.0** | **98.8** | 97.9 | **98.0** | 89.3 |
| STSN (ours) | **95.7** | 98.6 | 96.2 | 88.8 | 98.0 | **98.8** | 97.8 | **92.0** |

Table 2: Results on PGM.

| Model | Test Accuracy (%) | | |
|---|---|---|---|
| | Neutral | Interpolation | Extrapolation |
| CNN+MLP (Barrett et al., 2018) | 33.0 | - | - |
| CNN+LSTM (Barrett et al., 2018) | 35.8 | - | - |
| ResNet-50 (Barrett et al., 2018) | 42.0 | - | - |
| Wild-ResNet (Barrett et al., 2018) | 48.0 | - | - |
| CoPINet (Zhang et al., 2019b) | 56.4 | - | - |
| WReN ($\beta = 0$) (Barrett et al., 2018) | 62.6 | 64.4 | 17.2 |
| VAE-WReN (Steenbrugge et al., 2018) | 64.2 | - | - |
| MXGNet ($\beta = 0$) (Wang et al., 2020) | 66.7 | 65.4 | 18.9 |
| LEN ($\beta = 0$) (Zheng et al., 2019) | 68.1 | - | - |
| DCNet (Zhuo & Kankanhalli, 2022) | 68.6 | 59.7 | 17.8 |
| T-LEN ($\beta = 0$) (Zheng et al., 2019) | 70.3 | - | - |
| SRAN (Hu et al., 2021) | 71.3 | - | - |
| Rel-Base (Spratley et al., 2020) | 85.5 | - | **22.1** |
| SCL (Wu et al., 2020) | 88.9 | - | - |
| MRNet (Benny et al., 2021) | 93.4 | 68.1 | 19.2 |
| MLRN (Jahrens & Martinetz, 2020) | 98.0 | 57.8 | 14.9 |
| STSN (ours) | **98.2** | **78.5** | 20.4 |

struction for both the neutral and extrapolation training sets (thus giving these components of the model exposure to a broader range of shapes and feature values), while the transformer reasoning module was trained on the primary task using only the extrapolation training set.

## 4.4 RESULTS

Table 3: Results on CLEVR-Matrices.

| Model | Test Accuracy (%) | | | |
|---|---|---|---|---|
| | Average | Logic | Location | Count |
| MLRN (Jahrens & Martinetz, 2020) | 30.8 | 47.4 | 21.4 | 23.6 |
| SCL (Wu et al., 2020) | 70.5 | 80.9 | 65.8 | 64.9 |
| STSN (ours) | **99.6** | **99.2** | **100.0** | **99.6** |

Table 1 shows the results on the I-RAVEN dataset. STSN achieved state-of-the-art accuracy when averaging across all configurations (95.7%), and on two out of seven configurations ('U-D' and 'O-

Table 4: Ablation study on the I-RAVEN dataset.

| Model | Test Accuracy (%) | | | | | | | |
| --- | --- | --- | --- | --- | --- | --- | --- | --- |
| | Average | Center | 2Grid | 3Grid | L-R | U-D | O-IC | O-IG |
| STSN | **95.7** | **98.6** | **96.2** | **88.8** | **98.0** | **98.8** | **97.8** | **92.0** |
| -dropout | 93.4 | 97.8 | 92.5 | 84.7 | 96.4 | 96.7 | 96.5 | 89.6 |
| -dropout, -slot attention | 71.0 | 90.0 | 71.0 | 59.4 | 73.8 | 75.4 | 74.5 | 53.0 |
| -dropout, -TCN | 86.5 | 97.0 | 76.0 | 69.2 | 96.0 | 96.0 | 95.6 | 75.8 |
| -dropout, $L = 4$ | 88.6 | 96.4 | 85.8 | 74.4 | 94.6 | 95.0 | 94.5 | 79.2 |
| -dropout, -augmentations | 90.3 | 96.1 | 88.3 | 80.8 | 93.2 | 93.7 | 94.8 | 85.2 |

IG'). The most notable improvement was on the 'O-IG' configuration (a large outer object surrounding an inner grid of smaller objects), probably due to the need for more flexible object-encoding mechanisms in this configuration. For PGM (Table 2), STSN achieved state-of-the-art accuracy on the neutral (98.2%) and interpolation (78.5%) regimes, and achieved the second-best performance on the extrapolation regime (20.4% for STSN vs. 22.1% for Rel-Base). The next best model on I-RAVEN, SCL (95%) performed worse on PGM (88.9%), perhaps due to its more limited object-encoding methods (PGM includes a large number of spatially overlapping objects). We evaluated the next best model on PGM, MLRN (98%), on I-RAVEN (using code from the authors' publicly available respository), and found that it displayed very poor performance (29.8%), suggesting that some aspect of its architecture may be overfit to the PGM dataset. Thus, STSN achieved a $\sim 5\%$ increase in average performance across both of the two datasets relative to the next best overall model (97.0% average performance on PGM Neutral and I-RAVEN for STSN vs. 92.0% for SCL), despite incorporating fewer problem-specific inductive biases.

To further investigate the utility of STSN's object-centric encoding mechanism, we evaluated STSN, SCL, and MLRN on our newly developed CLEVR-Matrices dataset (Table 3). STSN displayed very strong performance (99.6% average test accuracy), whereas both SCL (70.5% average test accuracy) and MLRN (30.8% average test accuracy) performed considerably worse. This is likely due to the fact that these models lack a precise object-centric encoding mechanism, and were not able to cope with the increased visual complexity of this dataset.

Finally, we also evaluated both STSN and SCL on a dataset involving analogies between feature dimensions (e.g., a progression rule applied to color in one row, and size in another row) (Hill et al., 2019). STSN outperformed SCL on this dataset as well (Table 12), likely due to the fact that SCL assumes that rules will be applied independently within each feature dimension. This result highlights the limitation of employing inductive biases that are overly specific to certain datasets.

## 4.5 ABLATION STUDY

We analyzed the importance of the different components of STSN in ablation studies using the I-RAVEN dataset (Table 4). For I-RAVEN, our primary STSN implementation employed dropout, which we found yielded a modest improvement in generalization, but our ablation studies were performed without dropout. Thus, the relevant baseline for evaluating the isolated effect of each ablation is the version of STSN without dropout. First, we removed the slot attention module from STSN, by averaging the value embeddings from the input feature vectors over the image space (i.e., using only a single slot per panel). The average test accuracy decreased by more than 20%, suggesting that object-centric representations play a critical role in the model's performance. The effect was particularly pronounced in the 'O-IG' (a large outer object surrounding an inner grid of smaller objects) and '3Grid' (a $3 \times 3$ grid of objects) configurations, likely due to the large number of objects per panel in these problems. Next, we performed an ablation on TCN, resulting in a test accuracy decrease of around 7%, in line with previous findings demonstrating a role of TCN in improved generalization (Webb et al., 2020). We also performed an ablation on the size of the reasoning module, finding that a smaller transformer ($L = 4$ layers) did not perform as well. Finally, we performed an ablation on the image augmentations performed during training, resulting in a test accuracy decrease of more than 3%, suggesting that the augmentations also helped to improve generalization. Overall, these results show that the use of object-centric representations was the most important factor explaining STSN's performance on this task.

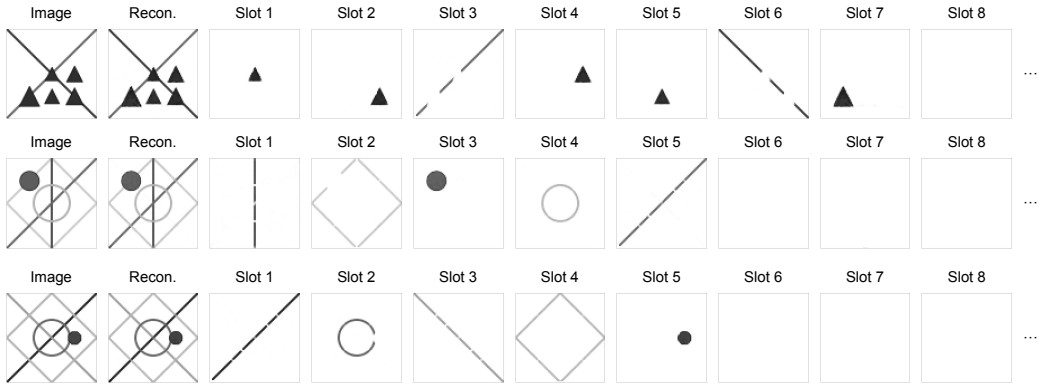

Figure 3: Slot-specific reconstructions generated by STSN. 3 problems were chosen at random from the PGM neutral test set. The first two images for each problem show the original image and the combined reconstruction. The following images show the slot-specific reconstruction for each of the slots. In general, STSN's slot attention module implemented a nearly perfect object-based segmentation of its input images, despite receiving no veridical segmentation information during training or test. STSN used 16 slots per image for this dataset, but generally left the slots not assigned to objects unused. Only 8 slots are pictured for these example problems since the remaining slot-specific reconstructions were completely blank.

## 4.6 VISUALIZATION OF OBJECT MASKS

We also visually inspected the attention behavior of STSN's slot attention module (Figure 3). We found that STSN's slot-specific reconstructions conformed nearly perfectly to the individual objects in the image panels of PGM, with the remaining slots left unused. This confirms that STSN was engaged in object-centric processing. We also evaluated STSN on I-RAVEN with a range of values for $\lambda$ (the parameter that governs the relative emphasis placed on the reconstruction loss), and found that with lower values of $\lambda$, STSN's reconstructions were no longer object-centric. With a value of $\lambda = 100$, STSN's reconstructions were blurrier, and multiple objects tended to be combined into a single slot (Figure 5 in Section A.4). With a value of $\lambda = 1$, STSN's reconstructions completely failed to capture the content of the original image (Figure 6). Interestingly, these changes in reconstruction quality were mirrored by changes in performance on the reasoning task, with an average test accuracy of $90.1\%$ for $\lambda = 100$ and $74.2\%$ for $\lambda = 1$ (relative to $95.7\%$ for $\lambda = 1000$, Figure 4). This is consistent with our hypothesis that encouraging high-quality reconstructions (through a sufficiently high weight on $\mathcal{L}_{recon}$) would encourage object-centric encoding behavior, which would in turn promote more generalizable visual reasoning strategies. Thus, for STSN to fully exploit its object-centric encoding mechanisms, it is important to use a high enough value of $\lambda$ so as to ensure high-quality reconstructions.

## 5 CONCLUSION AND FUTURE DIRECTIONS

We have presented a simple, general-purpose visual reasoning model, organized around the principle of object-centric processing. Our proposed model, STSN, displayed state-of-the-art performance on both of two challenging visual reasoning benchmarks, PGM and I-RAVEN, as well a novel reasoning benchmark with greater visual complexity, CLEVR-Matrices, despite the relative lack of problem-specific inductive biases. These results suggest that object-centric processing is a powerful inductive bias for abstract visual reasoning problems such as RPM.

Some previous work has proposed novel relational inductive biases for the purposes of achieving strong out-of-distribution generalization in visual reasoning problems (Webb et al., 2021; Zhang et al., 2021; Kerg et al., 2022). This work has often assumed (i.e., hand-coded) object-centric representations. We view our approach as complementary with these previous approaches, and suggest that a fruitful avenue for future work will be to pursue the integration of object-centric and relational inductive biases.

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

# A  APPENDIX

## A.1  CODE AND DATA AVAILABILITY

All code can be downloaded from https://github.com/Shanka123/STSN. The CLEVR-Matrices dataset can be downloaded from https://dataspace.princeton.edu/handle/88435/dsp01fq977z011.

## A.2  HYPERPARAMETERS AND TRAINING DETAILS

Images were resized to $80 \times 80$ for I-RAVEN and PGM, and $128 \times 128$ for CLEVR-Matrices. Pixels were normalized to the range $[0,1]$. For I-RAVEN, we also applied random online augmentations during training, including horizontal and vertical flips, rotations by multiples of $90°$, and brightness changes by a factor in the range [0.5,1.5]. Note that we applied the same augmentations to all the image panels in a given problem, so that the abstract rule remained the same.

Table 5 describes the hyperparameters for the convolutional encoder used on the I-RAVEN and PGM datasets, and Table 6 describes the hyperparameters used on the CLEVR-Matrices dataset. These encoders employed a positional embedding scheme consisting of 4 channels, each of which coded for the position of a pixel along one of the 4 cardinal directions (top→bottom, bottom→top, left→right, right→left), normalized to the range $[0, 1]$. These positional embeddings were projected through a fully-connected layer to match the number of channels in the convolutional feature maps, and then added to these feature maps. Feature maps were flattened along the spatial dimension, followed by layer normalization (Ba et al., 2016), and two 1D convolutional layers.

Table 5: CNN Encoder for I-RAVEN and PGM.

| Type | Channels | Activation | Kernel Size | Stride | Padding |
|---|---|---|---|---|---|
| 2D Conv | 32 | ReLU | $5 \times 5$ | 1 | 2 |
| 2D Conv | 32 | ReLU | $5 \times 5$ | 1 | 2 |
| 2D Conv | 32 | ReLU | $5 \times 5$ | 1 | 2 |
| 2D Conv | 32 | ReLU | $5 \times 5$ | 1 | 2 |
| Position Embedding | - | - | - | - | - |
| Flatten | - | - | - | - | - |
| Layer Norm | - | - | - | - | - |
| 1D Conv | 32 | ReLU | 1 | 1 | 0 |
| 1D Conv | 32 | - | 1 | 1 | 0 |

Table 6: CNN Encoder for CLEVR-Matrices.

| Type | Channels | Activation | Kernel Size | Stride | Padding |
|---|---|---|---|---|---|
| 2D Conv | 64 | ReLU | $5 \times 5$ | 1 | 2 |
| 2D Conv | 64 | ReLU | $5 \times 5$ | 1 | 2 |
| 2D Conv | 64 | ReLU | $5 \times 5$ | 1 | 2 |
| 2D Conv | 64 | ReLU | $5 \times 5$ | 1 | 2 |
| Position Embedding | - | - | - | - | - |
| Flatten | - | - | - | - | - |
| Layer Norm | - | - | - | - | - |
| 1D Conv | 64 | ReLU | 1 | 1 | 0 |
| 1D Conv | 64 | - | 1 | 1 | 0 |

For slot attention, we used $K = 9$ slots for I-RAVEN and CLEVR-Matrices, and $K = 16$ slots for PGM. The number of slot attention iterations was set to $T = 3$, and the dimensionality of the slots was set to $D_{slot} = 32$ for I-RAVEN and PGM, and $D_{slot} = 64$ for CLEVR-Matrices. The GRU had a hidden layer of size $D_{slot}$. The residual MLP had a single hidden layer with a ReLU activation, followed by a linear output layer, both of size $D_{slot}$.

Table 7 describes the hyperparameters for the slot decoder used on the I-RAVEN and PGM datasets, and Table 8 describes the hyperparameters used on the CLEVR-Matrices dataset. Each of the $K$ slots was passed through the decoder, yielding a slot-specific reconstructed image $\tilde{x}_k$ and mask $m_k$. Each slot was first converted to a feature map using a spatial broadcast operation Watters et al. (2019), in which the slot was tiled to match the spatial dimensions of the original input image ($80 \times 80$ for I-RAVEN and PGM, $128 \times 128$ for CLEVR-Matrices). Positional embeddings were then added (using the same positional embedding scheme as in the encoder), and the resulting feature map was passed through a series of convolutional layers. The output of the decoder had 2 channels for I-RAVEN and PGM, and 4 channels for CLEVR-Matrices. One of these channels corresponded to the mask $m_k$, and the other channels corresponded to the reconstructed image $\tilde{x}_k$ (grayscale for I-RAVEN and PGM, RGB for CLEVR-Matrices). The slot-specific reconstructions were combined by applying a softmax to the masks (over slots) and computing a weighted average of the reconstructions, weighted by the masks.

Table 7: Slot Decoder for I-RAVEN and PGM.

| Type | Channels | Activation | Kernel Size | Stride | Padding |
|---|---|---|---|---|---|
| Spatial Broadcast | - | - | - | - | - |
| Position Embedding | - | - | - | - | - |
| 2D Conv | 32 | ReLU | $5 \times 5$ | 1 | 2 |
| 2D Conv | 32 | ReLU | $5 \times 5$ | 1 | 2 |
| 2D Conv | 32 | ReLU | $5 \times 5$ | 1 | 2 |
| 2D Conv | 2 | - | $3 \times 3$ | 1 | 1 |

Table 8: Slot Decoder for CLEVR-Matrices.

| Type | Channels | Activation | Kernel Size | Stride | Padding |
|---|---|---|---|---|---|
| Spatial Broadcast | - | - | - | - | - |
| Position Embedding | - | - | - | - | - |
| 2D Conv | 64 | ReLU | $5 \times 5$ | 1 | 2 |
| 2D Conv | 64 | ReLU | $5 \times 5$ | 1 | 2 |
| 2D Conv | 64 | ReLU | $5 \times 5$ | 1 | 2 |
| 2D Conv | 64 | ReLU | $5 \times 5$ | 1 | 2 |
| 2D Conv | 64 | ReLU | $5 \times 5$ | 1 | 2 |
| 2D Conv | 4 | - | $3 \times 3$ | 1 | 1 |

Table 9 gives the hyperparameters for the transformer reasoning module, and Table 10 gives the training details for all datasets. We used a reconstruction loss weight of $\lambda = 1000$ for all datasets. We used the ADAM optimizer (Kingma & Ba, 2014) and all experiments were performed using the Pytorch library (Paszke et al., 2017). Hardware specifications are described in Table 11.

Table 9: Hyperparameters for Transformer Reasoning Module. $H$ is the number of heads, $L$ is the number of layers, $D_{head}$ is the dimensionality of each head, and $D_{MLP}$ is the dimensionality of the MLP hidden layer.

| | I-RAVEN | | PGM | | CLEVR-Matrices |
|---|---|---|---|---|---|
| | | Neutral | Interpolation | Extrapolation | |
| $H$ | 8 | 8 | 8 | 8 | 8 |
| $L$ | 6 | 24 | 24 | 6 | 24 |
| $D_{head}$ | 32 | 32 | 32 | 32 | 32 |
| $D_{MLP}$ | 512 | 512 | 512 | 512 | 512 |
| Dropout | 0.1 | 0 | 0 | 0 | 0 |

To make the comparison between STSN and SCL as fair as possible, we trained a version of SCL on I-RAVEN for 500 epochs using image augmentations (the same as used for STSN), TCN, and

Table 10: Training details for all datasets.

| | I-RAVEN | PGM | | | CLEVR-Matrices |
| | | Neutral | Interpolation | Extrapolation | |
|---|---|---|---|---|---|
| Batch size | 16 | 96 | 96 | 96 | 64 |
| Learning rate | $4e-4$ | $8e-5$ | $8e-5$ | $8e-5$ | $8e-5$ |
| LR warmup steps | 75k | 10k | 10k | 10k | 10k |
| Epochs | 500 | 161 | 83 | 71 | 200 |

Table 11: Hardware specifications for all datasets.

| | |
|---|---|
| I-RAVEN | 1 A100, 40GB RAM |
| PGM-Neutral | 6 A100, 40GB RAM |
| PGM-Interpolation | 6 A100, 40GB RAM |
| PGM-Extrapolation | 6 A100, 40GB RAM |
| CLEVR-Matrices | 8 A100, 80GB RAM |

dropout. TCN was applied following the last feedforward residual block of the first scattering transformation $\mathcal{N}^a$. Dropout with a probability of 0.1 was applied during training to all layers in the second scattering transformation $\mathcal{N}^r$.

We also compared STSN to a hybrid model that combined SCL and slot attention, termed 'Slot-SCL'. For this model, we replaced SCL's object and attribute transformations ($\mathcal{N}^o$ and $\mathcal{N}^a$) with the slot attention module, using the same hyperparameters as STSN. The outputs of slot attention were concatenated and passed to SCL's rule module ($\mathcal{N}^r$). This model also employed the same image augmentations, dropout, and TCN as our model. The model was trained for 400 epochs. The results for all comparisons with SCL on I-RAVEN reflect the best out of 5 trained models.

## A.3 ANALOGIES BETWEEN FEATURE DIMENSIONS

Table 12: Results on dataset involving analogies between feature dimensions (Hill et al., 2019) for *LABC* regime.

| | Test Accuracy (%) | | |
|---|---|---|---|
| Model | Average | Novel Domain Transfer | Novel Attribute Values (Extrapolation) |
| SCL (Wu et al., 2020) | 83.5 | 94.5 | 72.5 |
| STSN (ours) | **88** | **98.5** | **77.5** |

## A.4 EFFECT OF $\lambda$

Figures 4-6 show the effect of $\lambda$ (the hyperparameter governing the relative influence of the reconstruction loss) on reconstruction quality and object-centric processing.

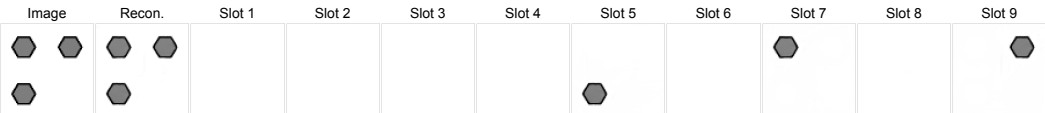

Figure 4: Slot-specific reconstructions generated by STSN for $\lambda = 1000$ on I-RAVEN.

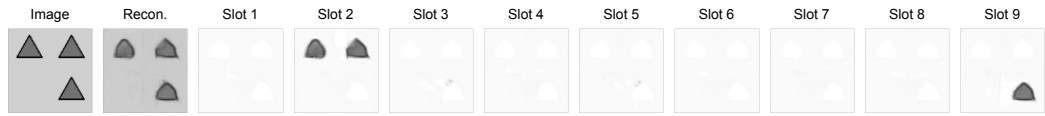

Figure 5: Slot-specific reconstructions generated by STSN for $\lambda = 100$ on I-RAVEN.

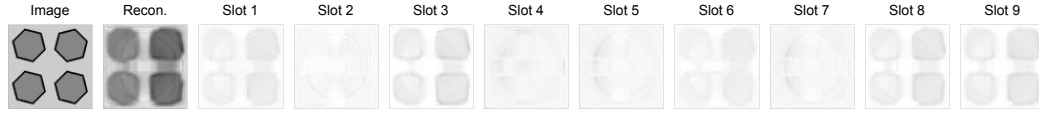

Figure 6: Slot-specific reconstructions generated by STSN for $\lambda = 1$ on I-RAVEN.

## A.5 CLEVR-MATRICES EXAMPLES

Figures 7-12 show some additional example problems from the CLEVR-Matrices dataset, along with annotations describing their problem type and the rules for each attribute.

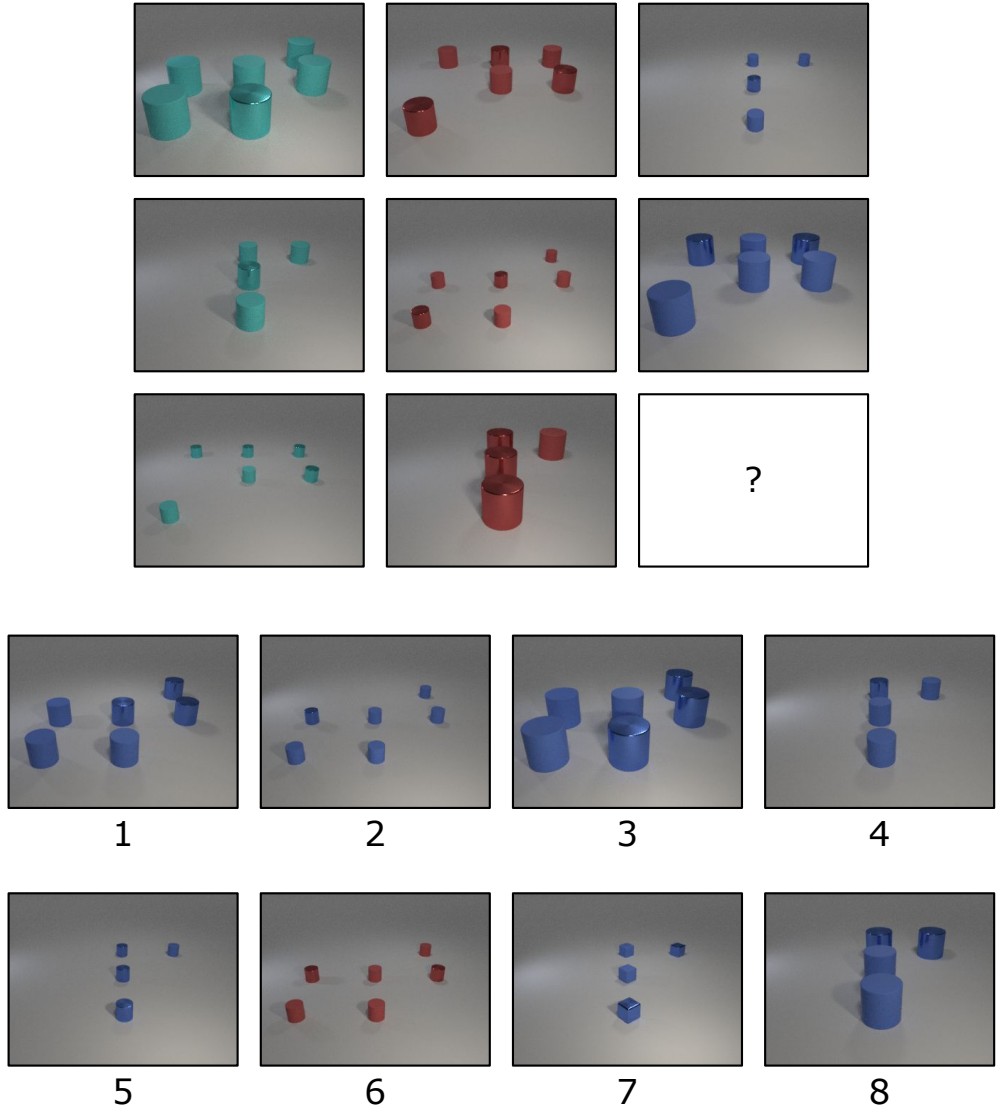

Figure 7: Problem type: location. Rules: [color: constant], [shape: null], [size: distribution-of-3], [location: distribution-of-3].

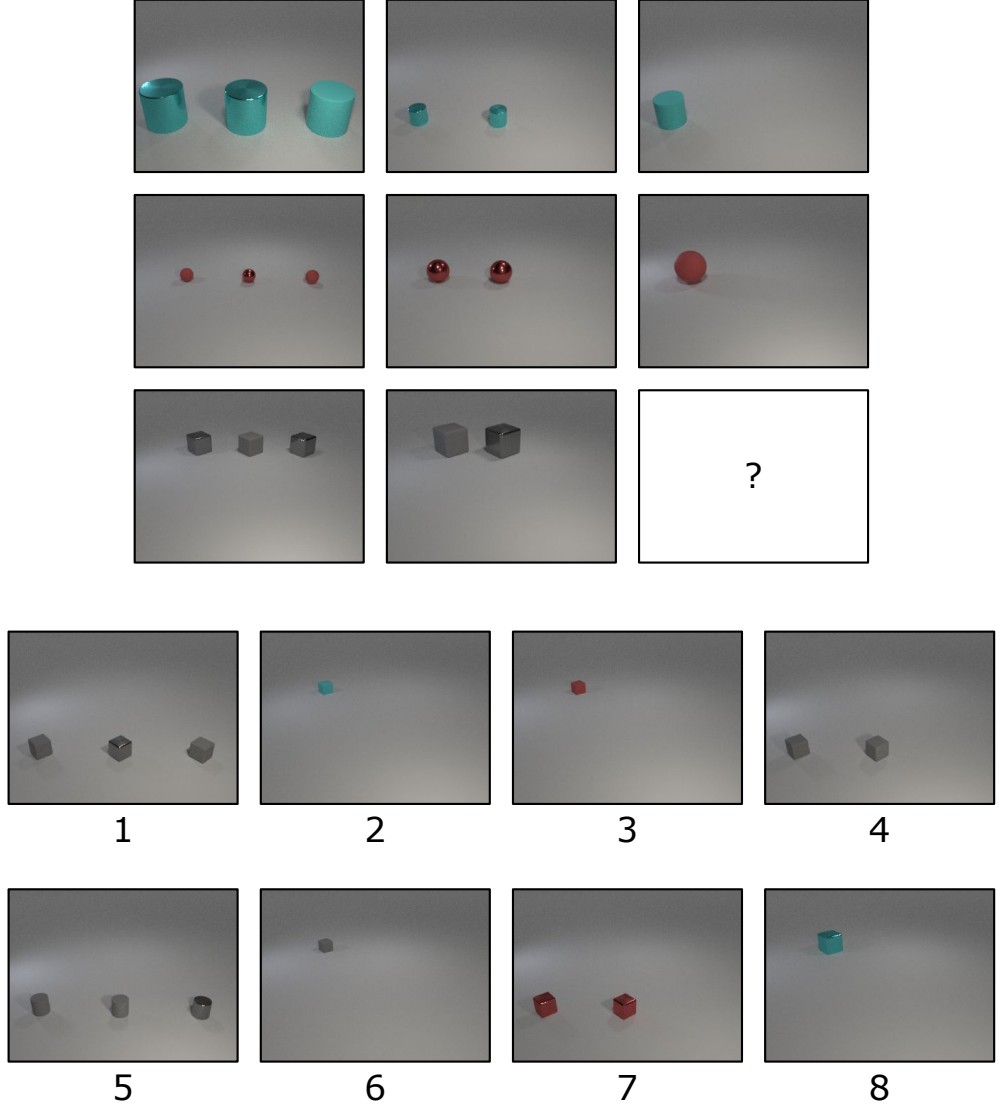

Figure 8: Problem type: location. Rules: [color: constant], [shape: constant], [size: distribution-of-3], [location: progression].

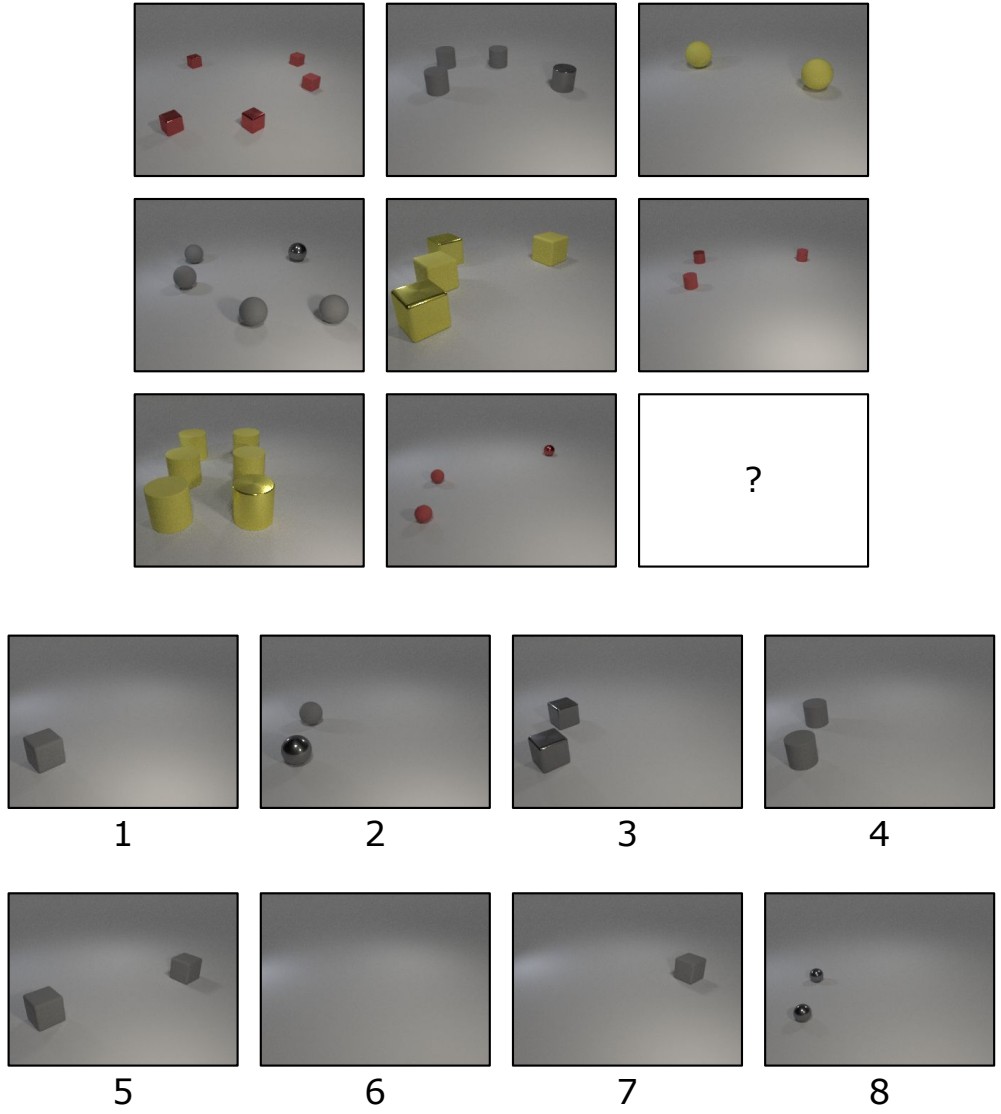

Figure 9: Problem type: logic. Rules: [color: distribution-of-3], [shape: distribution-of-3], [size: distribution-of-3], [location: AND].

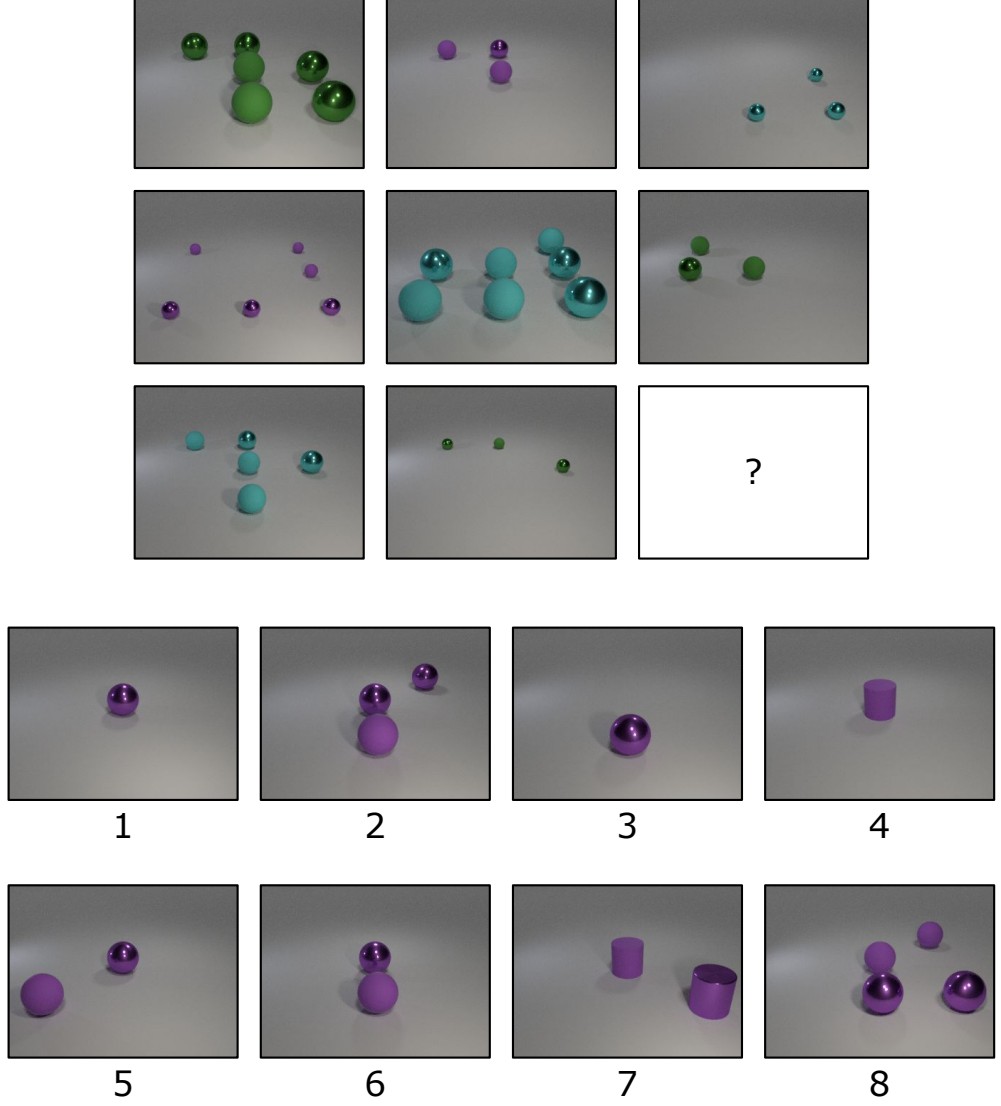

Figure 10: Problem type: logic. Rules: [color: distribution-of-3], [shape: null], [size: distribution-of-3], [location: XOR].

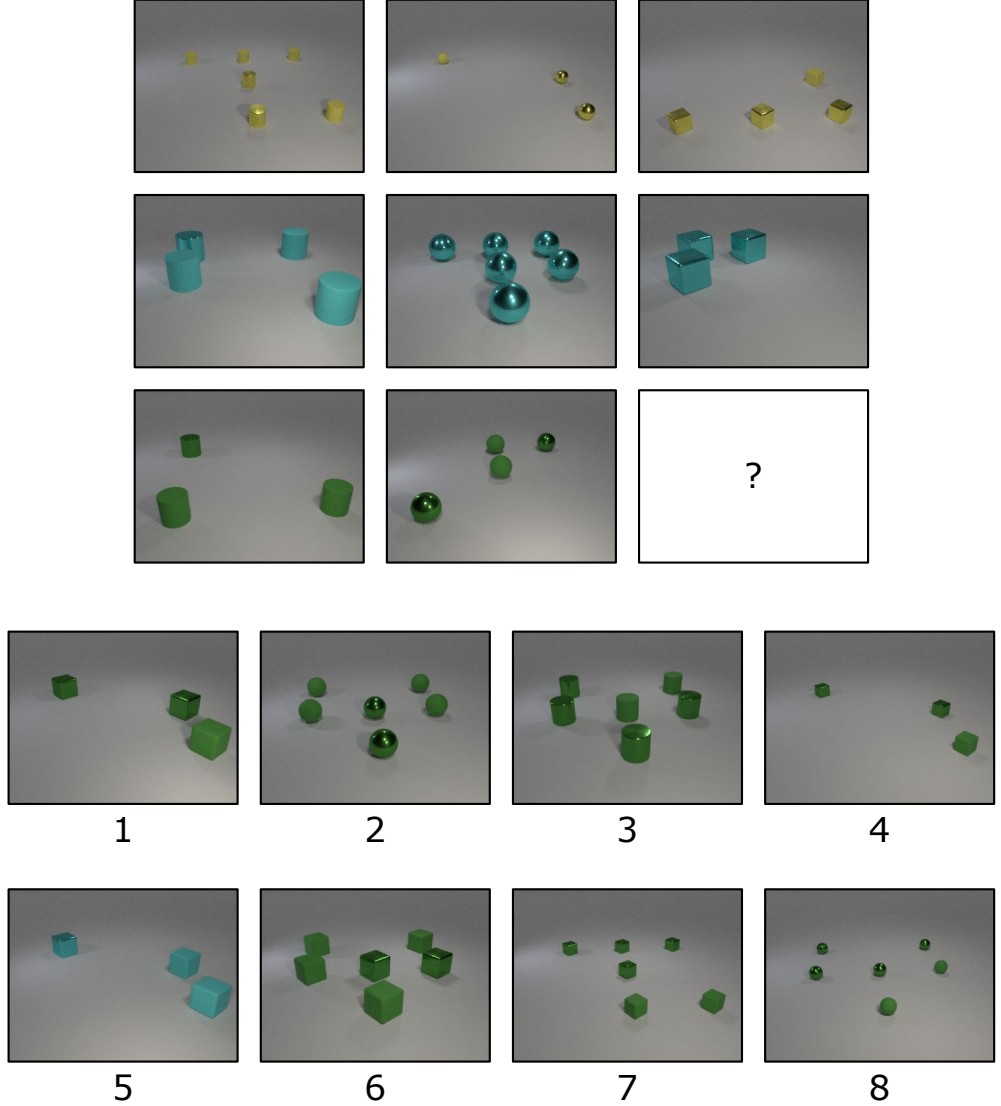

Figure 11: Problem type: count. Rules: [color: constant], [shape: constant], [size: constant], [count: distribution-of-3].

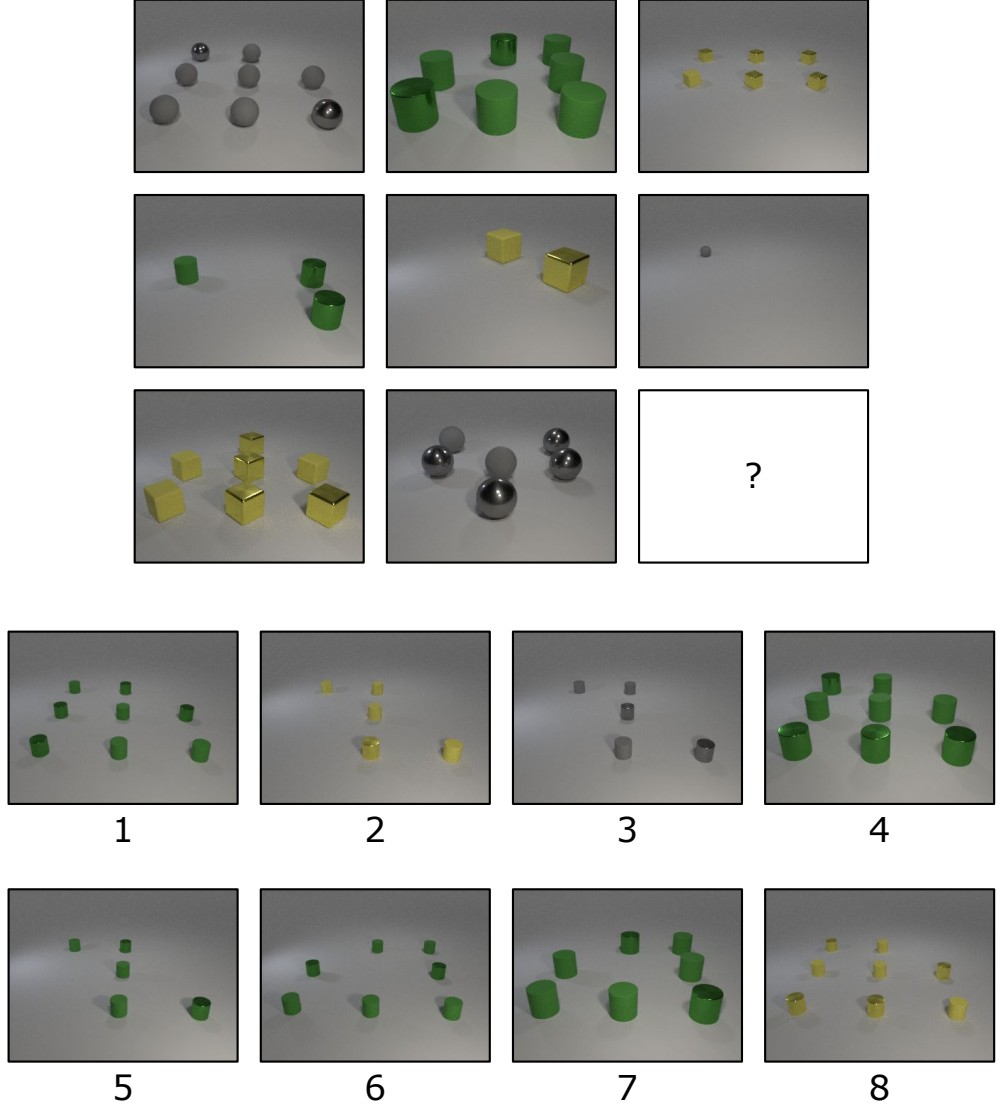

Figure 12: Problem type: count. Rules: [color: distribution-of-3], [shape: distribution-of-3], [size: distribution-of-3], [count: progression].

