# OpenReview forum: "Learning to reason over visual objects"
_ICLR.cc/2023/Conference — ICLR 2023 poster_

### Official Review · Reviewer_7pmp · 2022-10-21

**Confidence:** 4
**Correctness:** 3
**Technical Novelty And Significance:** 3
**Empirical Novelty And Significance:** 2
**Recommendation:** 6

**Clarity, Quality, Novelty And Reproducibility:**

The paper presentation is clear. The experimental designs and results are reasonable.

**Strength And Weaknesses:**

The proposed framework is straightforward and is well described in the paper. I personally very much like the simplicity of the proposed method. Tables 1 and 2 seem to be comprehensive lists of recent works on these tasks.

Despite the fact that I like the overall idea of the paper (it is simple and improves over the baselines). However, there are still a few weaknesses in the current version of the paper.

The first question is at the conceptual level. So this paper claims that “suggesting that an inductive bias for object-centric processing is a key component of visual reasoning” in the intro. I think this is one of the most important contribution of the paper (testing object-centric inductive bias in visual reasoning). However, this inductive bias has been tested many times in similar contexts, including the ALOE paper this paper cited. Therefore, this paper has a relatively low scientific contribution.

The second comment comes with my first one. If we look at the model performances, this paper does not achieve state-of-the-art performance on the testing tasks. I understand that this is because of the simplicity of the framework / not using any problem-specific inductive biases. But this makes the contributions poorly supported. One idea is to test the proposed framework on other tasks.

Regarding the non-SOTA performance on these benchmarks, I am wondering if this is because the transformer architecture is not designed for perform "reasoning" on this task. Specifically, if you look at the SCL paper from Wu et al, a major claim they are trying to make is that RPM-style tasks require reasoning about "the meta-level relationship" between concepts: understanding "progression," no matter whether it is applied to the axis of object colors or object shapes. However, Transofrmers (and other GNNs) only have object-level parameter sharing, but not factor-level parameter sharing. I wonder if this could be the reason why it underperforms several baselines.

There are a few comments about the experimental design:

1. Have you tried to train SlotAttention alone and just fix the encoding while training your transformer?
2. What's the conclusitve message from Section 4.6? Should we always use a high lamdba?

A few writing suggestions:

1. I think the title is a bit too broad. You probably want to be a little more specific.
2. The first paragraph in the intro does not have any supportive papers.
3. Given the length of the paper, I think the authors can expand their description of the Slot Attention module a bit more. This will help make the paper more self-contained.
4. Even though your model is not the state-of-the-art performer, you should still highlight the best-performing model in the tables.

**Summary Of The Paper:**

This paper presents a framework based on Slot Attention (for object-centric representation learning) and Transformer (for reasoning) to tackle the task of RAVEN's Progression Matrices. The input to the network is composed of 8 images as the context, and 8 images as candidate answers to be filled in as the ninth image. These images are generated synthetically, composed of geometric shapes of primitive shapes, sizes, colors, etc. The encoder module encodes each image into a set of slots, each of which corresponds to an "object" in the image and uses a transformer to rank all answers based on the slot-based representations.

**Summary Of The Review:**

While overall I very much like the simplicity of the proposed framework, there are a few weaknesses that have been listed. I think the current version of the paper is not ready for publish at ICLR.

---

> ### Author Response · Authors · 2022-11-19
> **Reply to reviewer 7pmp**
>
> We thank the reviewer for their helpful suggestions and comments. We have done our best to address each of them in our revised manuscript. We ask that the reviewer see our general overview above for a summary of the major additions to the manuscript (SOTA performance on both PGM and I-RAVEN, SOTA performance on a newly proposed CLEVR Matrices dataset). Below, we give a point-by-point reply to each of the reviewer’s specific concerns.
>
> -*So this paper claims that “suggesting that an inductive bias for object-centric processing is a key component of visual reasoning” in the intro. I think this is one of the most important contribution of the paper (testing object-centric inductive bias in visual reasoning). However, this inductive bias has been tested many times in similar contexts, including the ALOE paper this paper cited. Therefore, this paper has a relatively low scientific contribution.*
>
> We agree that our initial manuscript did not adequately explain what exactly the scientific contribution of our work is. It is true that our proposed method does not involve any fundamentally new components, and it is also true that the ALOE model, which has a similar structure at a high level, was previously evaluated on other visual reasoning tasks. However, a key issue evaluated in the present work, that has not, to our knowledge, been previously investigated, is the question of whether such object-centric processing will also facilitate *abstract* visual reasoning (i.e. visual analogy, as opposed to concrete visual reasoning tasks such as question answering from video). All of the best performing methods on the tasks that we investigate incorporated highly task-specific inductive biases, and it was therefore quite unclear whether a more general-purpose model might perform well on these tasks given access to object-centric representations, or whether these task-specific inductive biases were really needed. Our most recent results demonstrate that object-centric processing is extremely useful on abstract visual reasoning tasks, and indeed even appears to be more important than any of the previously proposed task-specific mechanisms. We have revised the manuscript to clarify these aspects of our contribution.
>
> -*If we look at the model performances, this paper does not achieve state-of-the-art performance on the testing tasks. I understand that this is because of the simplicity of the framework / not using any problem-specific inductive biases. But this makes the contributions poorly supported.*
>
> This is absolutely a reasonable concern given the results in our initial submission. However, as described in our general rebuttal above, our method now obtains state-of-the-art results on both the PGM and I-RAVEN datasets.
>
> -*One idea is to test the proposed framework on other tasks.*
>
> This is an excellent suggestion. To address this, we developed a new dataset, the CLEVR Matrices, that has a similar RPM-like task structure as the PGM and RAVEN datasets, but is characterized by significantly greater visual complexity. Our method achieves very strong results on this task (99.6% test accuracy), whereas the best previous model (SCL) performs poorly (70.5% test accuracy), presumably due to its limited object encoding mechanisms.
>
> -*Regarding the non-SOTA performance on these benchmarks, I am wondering if this is because the transformer architecture is not designed for perform "reasoning" on this task. Specifically, if you look at the SCL paper from Wu et al, a major claim they are trying to make is that RPM-style tasks require reasoning about "the meta-level relationship" between concepts: understanding "progression," no matter whether it is applied to the axis of object colors or object shapes. However, Transformers (and other GNNs) only have object-level parameter sharing, but not factor-level parameter sharing. I wonder if this could be the reason why it underperforms several baselines.*
>
> This is a very interesting point. It is indeed the case that SCL makes the assumption that abstract rules (e.g. progression) will be independently applied to each feature dimension, and that the structure of the rules will also be shared between feature dimensions. Surprisingly, our results show that accurate object-centric processing is more important than this particular inductive bias. An interesting open question is whether the object-centric inductive bias in our model can be combined with this key inductive bias from SCL (sharing of parameters between feature dimensions), by, for instance, sharing parameters between transformer heads.
>
> -*Have you tried to train SlotAttention alone and just fix the encoding while training your transformer?*
>
> This is a good question. We initially thought that this might help, but found in experiments on I-RAVEN that using a pretrained, frozen slot attention did not provide any clear advantage (though it also didn’t provide a clear disadvantage).

---

> > ### Author Response · Authors · 2022-11-19
> > **Reply to reviewer 7pmp (continued)**
> >
> > -*What's the conclusive message from Section 4.6? Should we always use a high lamdba?*
> >
> > We thank the reviewer for bringing this issue to our attention. The conclusive message of this section is that it is important to use a high enough value of $\lambda$ to encourage high quality reconstructions. The need to reconstruct the images provides an important pressure to learn more general purpose visual representations, which in turn provides pressure to learn object-centric representations (since this is an easier way to represent complex visual scenes). Without this pressure, the model does not learn object-centric representations (with $\lambda$=1 or $\lambda$=100), and generalizes poorly on the reasoning task. With a higher value of $\lambda$ ($\lambda$=1000), the model learns object-centric representations, produces high-quality reconstructions, and generalizes well on the reasoning task. Thus, the reconstruction objective acts as a regularization term, forcing the model to learn more general-purpose, object-centric representations, which in turn improves reasoning. We have added test results on I-RAVEN for the model with lower values of $\lambda$, and also added a discussion of these issues to section 4.6.
> >
> > -*I think the title is a bit too broad. You probably want to be a little more specific.*
> >
> > We thank the reviewer for this suggestion. Unfortunately, it does not appear possible to change the title of the submission at this stage.
> >
> > -*The first paragraph in the intro does not have any supportive papers.*
> >
> > We agree that some additional references would be helpful here. We have added a reference to a useful review of analogy in human reasoning [1], and a reference for the specific claim that RPM (Raven’s Progressive Matrices) problems are uniquely diagnostic of reasoning ability [2].
> >
> > [1] Holyoak, K. J. (2012). Analogy and relational reasoning.
> >
> > [2] Snow, R. E., Kyllonen, P. C., & Marshalek, B. (1984). The topography of ability and learning correlations. Advances in the psychology of human intelligence, 2(S 47), 103.
> >
> > -*Given the length of the paper, I think the authors can expand their description of the Slot Attention module a bit more. This will help make the paper more self-contained.*
> >
> > This is an excellent suggestion. We have now added an expanded explanation of slot attention (section 3.2).
> >
> > -*Even though your model is not the state-of-the-art performer, you should still highlight the best-performing model in the tables.*
> >
> > We have now made this change.

---

> ### Comment · Reviewer_7pmp · 2022-11-24
> **Thanks for your response.**
>
> Thank you for your response. The response and the revisions to the paper (including the new experiments) have greatly addressed my concerns. I am happy to raise my score. Speaking of the paper writing, I think the authors should cite more cognitive-science work in their first two paragraphs when they make claims such as "Human reasoning is driven by a capacity to extract simple, low-dimensional abstractions from complex, high-dimensional inputs." Also, the authors should add clarifications about their "focus/contribution" to the introduction (studying object-centric representations for "abstract" reasoning). Meanwhile, I strongly encourage authors to update their title to be more specific.
>
> I am giving 6 (instead of 8) primarily due to the writing of the paper, and overall I vote for acceptance.

---

> > ### Author Response · Authors · 2022-12-01
> > **Followup**
> >
> > We would like to thank the reviewer for addressing our revisions and updating their score. We also appreciate their continued advice regarding the writing. We are not able to upload a revision at this stage, but, if the paper is accepted, we will have an opportunity to upload a final 'camera-ready' version. We intend to address each of the remaining concerns in that revision. Below, we provide a point-by-point description of the planned revisions:
> >
> > ### Additional references in first paragraph:
> >
> > We agree that some additional references would be useful here. The first sentence (‘Human reasoning is driven by a capacity to extract simple, low-dimensional abstractions from complex, high-dimensional inputs.’) is elaborated in the second sentence, where the specific low-dimensional abstractions are articulated (‘...objects, relations, and higher-order patterns…’). We have added the following references to support these claims:
> >
> > Charles Spearman. The nature of "intelligence" and the principles of cognition. Macmillan, 1923.
> >
> > Mary L Gick and Keith J Holyoak. Schema induction and analogical transfer. Cognitive psychology, 15(1):1–38, 1983.
> >
> > Brenden M Lake, Tomer D Ullman, Joshua B Tenenbaum, and Samuel J Gershman. Building machines that learn and think like people. Behavioral and brain sciences, 40, 2017.
> >
> > We have also added an additional reference to the following sentence about the specific importance of analogy:
> >
> > Dedre Gentner. Structure-mapping: A theoretical framework for analogy. Cognitive science, 7(2): 155–170, 1983.
> >
> > Finally, we note that the other references in this paragraph add additional support regarding the importance of relations and higher-order patterns (both of which are central to analogy), and the references in the third paragraph provide support regarding the importance of objects.
> >
> > ### Summary of contributions:
> >
> > In order to emphasize our specific contributions (i.e. the focus on *abstract* visual reasoning), we have added an additional sentence to the final paragraph of the introduction. The first two sentences of that paragraph now read:
> >
> > **Recently, a number of methods have been proposed for the extraction of precise object-centric representations directly from pixel-level inputs, without the need for veridical segmentation data (Greff et al., 2019; Burgess et al., 2019; Locatello et al., 2020; Engelcke et al., 2021). While these methods have been shown to improve performance in some visual reasoning tasks, including question answering from video (Ding et al., 2021) and prediction of physical interactions from video (Wu et al., 2022), previous work has not addressed whether this approach is useful in the domain of *abstract* visual reasoning (i.e. visual analogy). To address this, we developed…**
> >
> > The second sentence of this paragraph is a new addition. The references are:
> >
> > David Ding, Felix Hill, Adam Santoro, Malcolm Reynolds, and Matt Botvinick. Attention over learned object embeddings enables complex visual reasoning. Advances in neural information processing systems, 34:9112–9124, 2021.
> >
> > Ziyi Wu, Nikita Dvornik, Klaus Greff, Jiaqi Xi, Thomas Kipf, and Animesh Garg. Slotformer: Long-term dynamic modeling in object-centric models. In UAI 2022 Workshop on Causal Representation Learning, 2022.
> >
> > ### Title:
> >
> > We agree that a more specific title would be better. We have reached out to the organizers in order to change the title in openreview, and we will change the title in the pdf version of our camera-ready paper if accepted. We plan to change the title to: “Abstract visual reasoning through learned object representations”

---

> > ### Author Response · Authors · 2022-12-09
> > **Followup**
> >
> > We would like to thank the reviewer once again for their continued engagement and helpful feedback. We would also like to inquire whether the proposed revisions to the paper satisfactorily address the reviewer's remaining concerns about the writing, or whether any additional changes are needed?

---

### Official Review · Reviewer_4JwD · 2022-10-24

**Confidence:** 3
**Correctness:** 3
**Technical Novelty And Significance:** 2
**Empirical Novelty And Significance:** 2
**Recommendation:** 6

**Clarity, Quality, Novelty And Reproducibility:**

Slot attention is based on prior work (Locatello et al.). This makes it seem like this work is an application of an existing method to this problem. The authors need to clarify the main technical contributions of this paper.

Technical details (section 3.2, 3.3) are only verbally described. Although I can get the gist of the method, the specific details need to be discussed with better clarity (e.g. mathematical expressions).

The reconstruction loss is vaguely described as the slot decoder in Locatello et al. Paper needs to include these details to be self contained.

It is hard to spot the best methods in the result tables, which makes it difficult to gauge the relative performance advantage of the proposed method.

There needs to be a detailed discussion about the baselines providing intuition about each method being compared against.

**Strength And Weaknesses:**

Pros
* Paper is easy to follow
* Hyperparameter choices are discussed
* Ablations show impact of each component
* Helpful visualization of the trained slot representations

Cons
* Scope of method is limited
* Limited technical novelty as the method is largestly based on prior work
* Experimental results are mixed and hard to contextualize due to lack of details about baselines
* Technical details are not described in detail

**Summary Of The Paper:**

The paper tackles RPM (Raven’s Progressive Matrices) problems where a set of context images are given and a continuation image is supposed to be selected from a given set of image choices. The images are governed by abstract rules which needs to be inferred by the method, which is claimed to parallel human reasoning capabilities. The authors propose a method which largely builds on the slot attention method of Locatello et al. On two RPM benchmarks (PGM, I-Raven), the authors show competitive performance compared to prior methods.

**Summary Of The Review:**

Raising my score to 6 post-rebuttal.

This paper builds on an existing approach designed for object centric processing and applies it to abstract reasoning problems involving structured image patterns. Although the problem and approach are interesting, the novel technical contributions of this work are unclear and the paper further has issues in the experimental setup (see detailed comments).

---

> ### Author Response · Authors · 2022-11-19
> **Reply to 4JwD**
>
> We thank the reviewer for their helpful suggestions and comments. We have done our best to address each of them in our revised manuscript. We ask that the reviewer see our general overview above for a summary of the major additions to the manuscript (SOTA performance on both PGM and I-RAVEN, SOTA performance on a newly proposed CLEVR Matrices dataset). Below, we give a point-by-point reply to each of the reviewer’s specific concerns.
>
> -*Scope of method is limited*
>
> We are not certain which aspect of the method the reviewer feels is limited. We argue that the method is actually quite general, and that is one of its major strengths. The only strong assumption made by the model is that visual inputs will be decomposable into objects, but this is arguably a highly general assumption, and likely to apply to any realistic visual task (perhaps excluding synthetic tasks designed to violate this property). The reasoning component, a generic transformer, preserves the natural permutation invariance of objects (not imposing any ordering), and models interactions between objects using a general-purpose attention mechanism that does not assume anything about the specific structure of the problem.
>
> Perhaps the reviewer means to highlight that our *evaluation* was limited. We previously evaluated our model only on the PGM and I-RAVEN datasets, both of which are based on highly synthetic, simplistic visual elements (2D grayscale geometric forms). To address this, and to highlight our proposed method’s generality, we developed a novel abstract reasoning benchmark, the CLEVR Matrices, characterized by significantly greater visual complexity. Our method significantly outperformed the previous best overall method on this task (99.6% accuracy for our method vs. 70.5% accuracy for SCL), emphasizing the general-purpose nature of our approach.
>
> -*Limited technical novelty as the method is largely based on prior work.*
> -*Slot attention is based on prior work (Locatello et al.). This makes it seem like this work is an application of an existing method to this problem. The authors need to clarify the main technical contributions of this paper.*
>
> The primary contribution of our work is to demonstrate that a general-purpose model, armed with an object-based visual representation, is capable of state-of-the-art abstract visual reasoning. To our knowledge, this has not been previously investigated, and constitutes a rather surprising result, given that the previous best performing methods have been specifically designed for these tasks. We do not claim to have proposed any fundamentally novel architectural components, but our view is that this is not always necessary for a paper to make a useful scientific contribution. In this case, we argue that our results constitute a significant scientific contribution, since they clearly demonstrate that object-centric representations are more important for abstract visual reasoning than any of the previously proposed, task-specific components. This aligns well with previous results in other visual task domains (e.g. question answering from video), but goes beyond them by showing that such object-centric representations are also critically important for *abstract* visual reasoning (i.e. visual analogy).
>
> -*Experimental results are mixed and hard to contextualize due to lack of details about baselines.*
>
> -*Technical details are not described in detail.*
>
> -*Technical details (section 3.2, 3.3) are only verbally described. Although I can get the gist of the method, the specific details need to be discussed with better clarity (e.g. mathematical expressions).*
>
> -*The reconstruction loss is vaguely described as the slot decoder in Locatello et al. Paper needs to include these details to be self contained.*
>
> -*There needs to be a detailed discussion about the baselines providing intuition about each method being compared against.*
>
> We thank the reviewer for these suggestions. We agree that these aspects were not adequately described in our initial submission, and have now revised the manuscript accordingly, adding more detail about our proposed method (sections 3.2-3.4), as well as more detail about the baselines we compare against (section 4.2).
>
> -*It is hard to spot the best methods in the result tables, which makes it difficult to gauge the relative performance advantage of the proposed method.*
>
> We have now used bold text to highlight the best results for each dataset.

---

> > ### Comment · Reviewer_4JwD · 2022-11-22
> > **Thank you for the response**
> >
> > In light of the author response and the new experiments I have a more favorable opinion of the paper. I am raising my score.

---

> > > ### Author Response · Authors · 2022-12-01
> > > **Followup**
> > >
> > > We thank the reviewer for addressing our revision and raising their score accordingly. We are pleased to see that the reviewer now has a more positive evaluation of the work. We would also like to inquire whether any of the concerns articulated in the initial review remain unaddressed, so that we might have the opportunity to address these remaining concerns.

---

### Official Review · Reviewer_hgxS · 2022-10-25

**Confidence:** 2
**Clarity, Quality, Novelty And Reproducibility:** Overall clarity, quality, and novelty…
**Correctness:** 3
**Technical Novelty And Significance:** 3
**Empirical Novelty And Significance:** 3
**Recommendation:** 6

**Strength And Weaknesses:**

*Strengths*

- The paper studies an interesting problem in training deep networks for challenging visual reasoning tests given to humans.
- Rather than having an overly hand-crafted solution that is not general, the proposed approach does seem to maintain good generality while still performing well on the task.
- The paper is written clearly.
- Ablations suggest all parts of the proposed model are important.

*Weaknesses*

- I'm not very familiar with this domain, so it is a bit difficult to evaluate the experiments. For example, it does seem like another approach SCL outperforms the method on both benchmarks. However, the authors claim that their proposed architecture is more general than that of SCL. The paper should do a better job of explaining exactly what ways the baselines which outperform their approach are less general, right now the claim is not well supported.

**Summary Of The Paper:**

The paper studies the problem of training deep networks to solve visual reasoning problems. It studies PGM and I-Ravens benchmarks, which pose challenging visual reasoning problems for humans, and explores if an end-to-end deep network can solve them.

The proposed method uses slot attention to individually encode each of the prompt images in an object-centric way (ideally separating the shapes in each prompt image) and combines them in another transformer to make the final prediction. The final training objective is reconstruction on each of the individual prompt images from the object-centric representation, and a classification objective for the task.

Overall the proposed method performs comparably to the best models on the benchmarks, while claiming to be more general.

**Summary Of The Review:**

The paper does seem to propose a novel approach for learning difficult visual reasoning tasks without too much hand-crafted structure. Results seem decent but are a bit hard to evaluate without detailed knowledge of the baselines, which perform better but the authors claim are less general. Overall I would defer to other reviewers who have more knowledge of this line of work.

---

> ### Author Response · Authors · 2022-11-19
> **Reply to reviewer hgxS**
>
> We thank the reviewer for their helpful suggestions and comments. We have done our best to address each of them in our revised manuscript. We ask that the reviewer see our general overview above for a summary of the major additions to the manuscript (SOTA performance on both PGM and I-RAVEN, SOTA performance on a newly proposed CLEVR Matrices dataset). Below, we give a point-by-point reply to each of the reviewer’s specific concerns.
>
> -*…it does seem like another approach SCL outperforms the method on both benchmarks.*
>
> Our updated results (described in our general rebuttal above, and in the paper) now show that our model obtains SOTA performance on both PGM (neutral) and I-RAVEN, as well as the Interpolation regime (out-of-distribution) of PGM. Additionally, our method outperforms SCL (99.6% vs. 70.5% test accuracy) on our newly proposed CLEVR Matrices task.
>
> -*The paper should do a better job of explaining exactly what ways the baselines which outperform their approach are less general, right now the claim is not well supported.*
>
> We agree that this was not adequately explained in our initial submission. The previous methods applied to these datasets are problem-specific in at least two major ways (which we have also attempted to clarify in the revised manuscript):
>
> 1. Many methods do not employ any object-centric processing at all, but the ones that do (e.g. SCL) employ a very limited form of object segmentation that is based on spatial location. For datasets such as PGM and RAVEN, this method offers a decent approximation, since objects often appear in a regular grid of locations. However, this approach will not work in general for more complex scenes with variable object location. Even for PGM and RAVEN, objects are sometimes overlapping, or located inside of other objects, which a location-based segmentation method cannot accommodate. In our proposed CLEVR Matrices problem set, we employed even more variable 3d object location (as is done in CLEVR), which poses an even greater challenge for location-based methods (illustrated by SCL’s poor performance on this dataset).
> 2. Most of the best performing previous methods explicitly build the structure of RPM problems into their architecture. For instance, SCL assumes that there will not be any interactions between feature dimensions (rules will be independently applied to each feature dimension). MRNet builds the row-wise and column-wise structure of RPM problems into its architecture. It is therefore not clear how these methods could even be applied to problems other than RPM. By contrast, our proposed method, STSN, contains only two highly general components, both of which have been independently applied to a range of previous tasks, and which, in combination, have been previously shown to perform well on other visual reasoning tasks (e.g. the performance of ALOE on question-answering from video).

---

> ### Author Response · Authors · 2022-12-01
> **Followup**
>
> We would like to thank the reviewer once again for their positive evaluation of our work, along with their helpful suggestions for improving the work. We have attempted to address these limitations in our revision. Specifically, we now show that our proposed approach achieves state-of-the-art performance on both PGM and I-RAVEN, along with a new task, CLEVR-Matrices. On this new task, by contrast, the previous best model, SCL, performs considerably worse (70.5% accuracy for SCL vs. 99.6% accuracy for our proposed method, STSN). This highlights the limitation of the task-specific object-encoding method employed by SCL. We have also added more description of the baselines that we compare against (section 4.2 of revision), and described their task-specific inductive biases (first two paragraphs of section 2). We hope that these revisions address the reviewer’s concerns.

---

### Official Review · Reviewer_j7eh · 2022-11-04

**Confidence:** 5
**Correctness:** 2
**Technical Novelty And Significance:** 2
**Empirical Novelty And Significance:** 2
**Recommendation:** 6

**Clarity, Quality, Novelty And Reproducibility:**

The model architecture is clear overall, but the concrete contributions need to be emphasized.

The novelty is low, the network architecture is a patching up of already exist models (slot attention, transformer, TCN) and does not achieve new SOTA. The importance of object-centric processing within visual reasoning is already well-known.

The results seem reproducible given the details in the text, but open-source would much increase reproducibility (or indicate the open source upon acceptance).

**Strength And Weaknesses:**

Pros:
1. It is valuable to show that object-centric models using modern modules (slot attention + transformer) can perform well on the RAVEN and PGM datasets.
2. It is informative to have ablation studies to show the improvement of each module in Table 3.

Cons:
1. It is unclear what contributions this work presents. The slot attention and transformer already exist, and this work piles them up.
2. This work claims STSN is a general-purpose architecture but do not show its performance over other visual reasoning tasks other than RPM tasks (like CLEVR [1]).
3. The model size is large compared to SCL. It is unclear whether the larger model size is a cost for general-purpose. Can you do ablation study over the size of the model (like the size of hidden dimension or the number of layers of Transformer)?

Minors:
* Abstract: "This work has ...  ", a bit confusing which work this part refers to, your work or the recent work in last sentence? (Also in Intro: "Much of this work ...")
* Sec 3.3 cnadidate -> candidate
* Figures 4 to 6 are not fit well into the page.
* The last page of the references needs re-format.

[1] Johnson, J., Hariharan, B., Van Der Maaten, L., Fei-Fei, L., Lawrence Zitnick, C., & Girshick, R. (2017). Clevr: A diagnostic dataset for compositional language and elementary visual reasoning. In Proceedings of the IEEE conference on computer vision and pattern recognition (pp. 2901-2910).

**Summary Of The Paper:**

This work finds a simple model (called STSN) consisting of slot attention and transformer modules. Without incorporating inductive biases specific to the RPM problem format, this model still displays near-SOTA performance on the I-RAVEN and PGM datasets. Such performances suggest the importance of object-centric processing within visual reasoning.

**Summary Of The Review:**

The contributions are not clear. As the model does not achieve new SOTA on both the I-RAVEN and PGM datasets, if the contribution goes to a general-purpose visual reasoning model, it should show its performance in other reasoning tasks that are related to objects.

---
Update (2022/11/22): raise from 3 to 5 after reading the rebuttal, comment below.

---
Update (2022/12/04): raise from 5 to 6. I hope the authors continue improving their paper (about clarity and rigorousness) and include the points they addressed in the rebuttal.

---

> ### Author Response · Authors · 2022-11-19
> **Reply to reviewer j7eh**
>
> We thank the reviewer for their helpful suggestions and comments. We have done our best to address each of them in our revised manuscript. We ask that the reviewer see our general overview above for a summary of the major additions to the manuscript (SOTA performance on both PGM and I-RAVEN, SOTA performance on a newly proposed CLEVR Matrices dataset). Below, we give a point-by-point reply to each of the reviewer’s specific concerns.
>
> -*It is unclear what contributions this work presents. The slot attention and transformer already exist, and this work piles them up.*
> -*The novelty is low, the network architecture is a patching up of already existing models (slot attention, transformer, TCN) and does not achieve new SOTA. The importance of object-centric processing within visual reasoning is already well-known.*
>
> First and foremost, the major contribution of our work is to show that object-centric processing enables state-of-the-art performance on both major abstract visual reasoning tasks, PGM and RAVEN, resulting in a general-purpose model that outperforms all previous approaches on these tasks, despite the lack of task-specific inductive biases. We do not claim to have proposed any fundamentally novel architectural components, but our view is that this is not always necessary for a paper to make a useful scientific contribution. It is true that object-centric processing has previously been shown to be useful in other visual reasoning tasks (e.g. question-answering from video and physical reasoning), but no previous work, to our knowledge, has investigated the extent to which this inductive bias is useful for *abstract* visual reasoning (i.e. visual analogy). Our results lead to the surprising conclusion that this inductive bias is even more important for abstract visual reasoning tasks than inductive biases that have been specifically designed for these tasks.
>
> -*This work claims STSN is a general-purpose architecture but do not show its performance over other visual reasoning tasks other than RPM tasks (like CLEVR [1]).*
>
> We agreed that this was a significant limitation of our initial submission, and therefore designed a new task, the CLEVR Matrices, to evaluate the generality of our proposed model. Our proposed method achieved very strong performance on this task (99.6% test accuracy), whereas the previous overall best method on PGM and RAVEN (SCL) performed much worse (70.5% test accuracy), likely due to the task-specific nature of the model’s object-encoding mechanisms.
>
> -*The model size is large compared to SCL. It is unclear whether the larger model size is a cost for general-purpose. Can you do ablation study over the size of the model (like the size of hidden dimension or the number of layers of Transformer)?*
>
> It is true that our model has more parameters than SCL. We investigated this issue, and found that a smaller model (transformer depth of $L=4$ vs. $L=6$ layers) still showed strong performance on this task (I-RAVEN test accuracy of 88.5% for the best performing model out of 5 runs), but did not perform as well as the standard version of our model (95.7% test accuracy), nor as well as the previously best performing model (95% test accuracy for SCL). Though it is possible that a more extensive hyperparameter search might yield different results, the results of this evaluation tentatively suggest that our model might indeed require more parameters. On balance, we think that this is a reasonable tradeoff, given that our model is significantly more general, and also displays the best performance across all three of the abstract visual reasoning tasks that we addressed. We have added a note reporting these results in the Appendix.
>
> We also thank the reviewer for their minor comments regarding clarity, typos, etc. We have addressed each of them in the revised manuscript. We would also like to note that we indeed intend to publicly release all code upon acceptance.

---

> > ### Comment · Reviewer_j7eh · 2022-11-23
> > **Thanks for the reply and revision**
> >
> > Thanks for your reply and revision.
> >
> > According to the revision (including the new I-RAVEN and PGM results, as well as the proposed CLEVR-Matrices dataset), I'm happy to increase my overall score from 3 to 5.
> >
> > However, I still have the following concerns (based on the current version):
> >
> > 1. The new results on I-RAVEN just outperform SCL by a small margin while using several tricks like dropout, data augmentation, and TCN (note that SCL did not use these). As indicated in Table 4, the performance of STSN would drop significantly without these tricks. It is reasonable when you focus on pursuing SOTA results. However, when the focus is testing the performance of a general-purpose architecture, it is better to keep these general tricks the same for all methods when possible.
> > 2. Good job showing the importance of slot attention and the object-centric inductive bias for visual reasoning tasks. However, as also pointed out by reviewer 7pmp, it is unclear how the Transformer compared to the SCL (therefore hard to judge your argument about general purpose vs. specific inductive bias). SCL could also benefit from the more advanced slot attention module for extracting object representations. Would STSN still outperform SCL with slot attention? When comparing two modules, it is more reasonable to control other factors to be the same as possible.
> > 3. Requiring fewer parameters is one of the advantages of adopting specific inductive bias. The tradeoff between the general purpose and the size of the model should be discussed in the main text (not only in the Appendix).
> > 4. Not sure whether you tune hyperparameters for both STSN and SCL for the CLEVR-Matrices dataset. I understand the time for rebuttal is limited, but if you tune hyperparameters for STSN, you should also do that for SCL for a fair comparison.
> >
> > Also, it needs to be corrected that SCL supports interactions between feature dimensions. This is explained in Appendix A of SCL:
> > > The reason _(for using a feedforward residual block)_ is to extract some potentially useful global information (such as "number" of objects).

---

> > > ### Author Response · Authors · 2022-12-01
> > > **Followup**
> > >
> > > We would like to thank the reviewer for their continued engagement, and for raising their score. We include below a point-by-point reply to the additional concerns raised here, including the results of a new experiment:
> > >
> > > ### Use of dropout, TCN, data augmentation
> > >
> > > We agree that it is important to control for these factors. Accordingly, we have now run an additional control experiment in which we trained a version of SCL on I-RAVEN that uses all of these same techniques. We trained 5 models for 500 epochs (the same amount of time used to train STSN on I-RAVEN, and 200 epochs longer than SCL was trained in the original work), and selected the model with the highest validation accuracy (the same model selection procedure used for STSN on I-RAVEN, and for the original SCL work). We used the same image augmentations as those used for STSN. In STSN, TCN was applied to the slots following slot attention, before being passed to the reasoning module. In SCL, the most analogous component to slot attention is the first scattering transformation $\mathcal{N}^{a}$, which is intended to extract the attributes for each object. Therefore, we applied TCN to the outputs of $\mathcal{N}^{a}$. In STSN, dropout was applied to each layer of the transformer reasoning module. The most analogous component in SCL is the second scattering transformation $\mathcal{N}^{r}$, which is intended to integrate information across panels and identify the rules for each feature dimension. Therefore, we applied dropout with a probability of 0.1 (same as in STSN) to each of the layers in $\mathcal{N}^{r}$.
> > >
> > > When combining these techniques with SCL, the best performing model achieved an average accuracy of 95.5% on I-RAVEN, which is still lower than the score of 95.7% achieved by STSN. Additionally, we point to SCL’s significantly worse performance on both PGM (88.9% for SCL vs. 98.2% for STSN) and CLEVR-Matrices (70.5% for SCL and 99.6% for STSN). We also would like to note that these techniques (image augmentations, TCN, dropout) can each be flexibly applied to a wide range of tasks, and thus do not undermine the status of STSN as a general-purpose model.
> > >
> > > If accepted, we will add these additional results to the camera-ready version of the paper.
> > >
> > > ### Combining slot attention with SCL
> > >
> > > We agree that it is interesting to consider how the inductive biases in SCL might be fruitfully combined with object-centric encoding methods such as slot attention. However, we would like to point out that SCL is not a module per se, but a fully-fledged architecture consisting of multiple modules, including modules for extracting objects and identifying their attributes ($\mathcal{N}^{o}$ and $\mathcal{N}^{a}$) and a module for identifying rules ($\mathcal{N}^{r}$). The most straightforward way to test this idea would be to replace both $\mathcal{N}^{o}$ and $\mathcal{N}^{a}$ with slot attention, but this would effectively be replacing half of the architecture. Furthermore, we doubt that this would work very well, since the assignment of objects to slots is random in slot attention (due to the permutation invariance of the slots), whereas the $\mathcal{N}^{r}$ transformation in SCL uses an MLP. This means that $\mathcal{N}^{r}$ would have to learn this permutation invariance from scratch, whereas the transformer reasoning module in STSN is naturally permutation invariant and thus works very well with slot attention.
> > >
> > > ### Adding the model size ablation to the main text
> > >
> > > We agree, and will add these results to the main text of the camera-ready paper, in the section describing ablation results (4.5).
> > >
> > > ### Hyperparameter tuning for CLEVR-Matrices
> > >
> > > We did not perform any hyperparameter tuning for either STSN or SCL on CLEVR-Matrices. The only change we made to STSN, relative to the implementation used on other tasks, is to change the encoder and decoder to the default encoder and decoder used on the CLEVR dataset in the original slot attention paper. This is the only version of STSN that we ran on CLEVR-Matrices.
> > >
> > > ### Interaction between feature dimensions in SCL
> > >
> > > We thank the reviewer for noting this detail about SCL. However, we would like to point out that this does not contradict the description of SCL in our paper. Specifically, in the paper we state that SCL 'assumes that rules are independently applied in each feature dimension'. According to Appendix A in the SCL paper, this feedforward residual block is only applied following the $\mathcal{N}^{a}$ transformation, and thus only considers global information at the *within-panel* level (i.e. the number of objects in a panel), as opposed to the *between-panel* interactions that are critical for identifying rules.

---

> > > ### Author Response · Authors · 2022-12-11
> > > **Additional experiment**
> > >
> > > We would like to thank the reviewer for raising their score. We would also like to report that we have now performed the additional experiment suggested by the reviewer, combining SCL and slot attention. Specifically, we replaced $\mathcal{N^{o}}$ and $\mathcal{N^{a}}$ with the same slot attention module used by STSN, feeding the outputs of slot attention to SCL’s $\mathcal{N^{r}}$ module. We also employed TCN, dropout, and image augmentations, as used in the best performing version of STSN. As predicted, this does not perform as well as the standard version of SCL, with the best model (out of 5 trained models) achieving an average test accuracy of 90.4% on I-RAVEN (as opposed to 95% for the original SCL model, and 95.5% for SCL combined with TCN, dropout, and image augmentations). This is most likely because slot attention randomly assigns objects to slots, and SCL’s MLP-based architecture must then learn a permutation-invariant reasoning procedure from scratch, whereas the transformer employed by STSN is permutation-invariant by design. We plan to add these results to the paper, and thank the reviewer for suggesting this experiment.

---

### Author Response · Authors · 2022-11-19
**General overview of revision**

We thank the reviewers for their insightful feedback and suggestions. We have now uploaded a substantially revised manuscript. This includes two very significant updates:

1. Our proposed model (STSN) now achieves **state-of-the-art performance on both the PGM and I-RAVEN datasets**. On PGM, this result (STSN test accuracy of 98.2%) was achieved by allowing the model to train for longer (the model had not fully converged on the training set at the time of our initial submission, as noted in the appendix of the original submission). Note that the criteria used to select an epoch for testing remained the same (the epoch with the best validation accuracy was selected for evaluation on the test set). On I-RAVEN, this result (STSN test accuracy of 95.7%) was achieved by adding dropout to the model (again using validation accuracy for model selection). We also include results for the PGM ‘Interpolation’ regime (an out-of-distribution test regime, in which an entire set of feature values is withheld during training and then included in test), where our model also achieves a new state-of-the-art score (STSN test accuracy of 78.5% vs. previous SOTA of 68.1%).
2. We have developed a **new abstract visual reasoning task, the CLEVR Matrices**, that employs a similar RPM-like problem structure, but with much greater visual complexity (similar to the CLEVR dataset, i.e. involving photorealistic rendered 3D objects). Our model also shows strong performance on this task (STSN test accuracy of 99.6%) with a relatively small training set (48k problems), whereas the previous best overall model (on PGM and RAVEN) performs significantly worse on this more challenging task (SCL test accuracy of 70.5%).

Thus, our model is now the **best-performing model on both major extant abstract visual reasoning tasks, as well as a newer, more challenging, visual reasoning task**, despite the fact that it does not employ any *task-specific* inductive biases. We take this as strong evidence that our model’s key inductive bias, object-centric visual processing, is a critically important element for abstract visual reasoning.

A concern expressed by many of the reviewers was a lack of novelty in our proposed method, and a lack of clarity regarding the scientific contribution of the work. We do not claim to have proposed any fundamentally novel architectural components, but our view is that this is not always necessary for a paper to make a useful scientific contribution. Rather, our goal in this project was to evaluate the extent to which two *general-purpose* components – object-centric visual processing, and transformer-style, attention-based reasoning mechanisms – might on their own (i.e., without any task-specific enhancements) enable abstract visual reasoning, and how such a general-purpose architecture might compare to more task-specific architectures. Previous work has demonstrated the value of these components in other domains. Most notably, the previously proposed ALOE model (which also combines object-centric processing and a transformer reasoning module) has demonstrated strong performance on certain forms of visual reasoning tasks, such as question-answering from video. Our results go beyond this in an important way, by showing that such a general-purpose architecture excels not only on relatively concrete visual reasoning tasks, such as question answering, but considerably more challenging and *abstract* tasks involving analogical reasoning (i.e., higher-order relations) such as we tested here.

---

> ### Author Response · Authors · 2022-11-19
> **General overview (continued)**
>
> The reviewers also raised concerns about a lack of clarity regarding the problem-specific inductive biases employed by previous methods. The previous methods applied to these datasets are problem-specific in at least two major ways (that we have also attempted to clarify in the revised manuscript):
>
> 1. Many methods do not employ any object-centric processing at all, but the ones that do (e.g. SCL) employ a very limited form of object segmentation that is based on fixed spatial location. For datasets such as PGM and RAVEN, this method offers a decent approximation, since objects often appear in a regular grid of locations. However, this approach will not work in general for more complex scenes with variable object location. Even for PGM and RAVEN, objects are sometimes overlapping, or located inside of other objects, which a location-based segmentation method cannot accommodate. In our proposed CLEVR Matrices problem set, we employed even more variable 3d object location (as is done in CLEVR), which poses an even greater challenge for location-based methods (illustrated by SCL’s poor performance on this dataset).
> 2. Most of the best performing previous methods explicitly, and more specifically build the structure of RPM problems into their architecture. For instance, SCL assumes that there will not be any interactions between feature dimensions (rules will be independently applied to each feature dimension). MRNet builds the row-wise and column-wise structure of RPM problems into its architecture. It is therefore not clear how these methods could even be applied to problems other than RPM. By contrast, our proposed method, STSN, contains only two highly general components, both of which have been independently applied to a range of previous tasks, and which, in combination, have been previously shown to perform well on other visual reasoning tasks (e.g. the performance of ALOE on question-answering from video).
>
> Another concern raised by reviewers was that we highlight the general-purpose nature of our proposed approach, but only test it on two highly similar datasets, PGM and I-RAVEN. We hope that our proposal of an entirely new dataset, the CLEVR Matrices, along with results showing our model’s strong performance on this dataset, helps to address this concern.
>
> The reviewers also noted the need for further clarification or explanation in some places, that we have tried to address in our revision. Below, we include specific point-by-point replies to each of the individual reviews.

---

### Decision · Program_Chairs · 2023-01-20

**Decision:**

Accept: poster

**Justification For Why Not Higher Score:**

While I recommend acceptance ultimately, to recommend as a spotlight or oral definitely seems a big stretch, due to limited technical novelty.

**Justification For Why Not Lower Score:**

I do think that the paper, while not strong technically, does ultimately have a contribution in the senses mentioned above. The SOTA results may well be surpassed soon enough, but the field has progressed not only from distinct jumps, but also small and continuous improvements.

**Metareview: Summary, Strengths And Weaknesses:**

This paper is premised on investigating if a combination of existing generic techniques / components, namely slot attention and transformer modules, are able to do well on abstract visual reasoning, specifically the Raven's Progressive Matrics (RPM) task. This model, STSN (Slot Transformer Scoring Network), does not have any task-specific inductive biases, yet does well on two standard datasets (I-RAVEN and PGM) compared to existing methods, as well as a new proposed dataset (CLEVR-Matrices) purportedly with greater visual complexity. The key claimed contribution is the demonstration that a generic inductive bias (object-centric processing) is able to tackle RPMs well, without necessitating problem-specific inductive biases.


-- STRENGTHS --

1) SOTA performance on multiple datasets, including 2 standard ones and 1 new proposed dataset

2) Method is simple and intuitive


-- WEAKNESSES --

1) Limited technical novelty: straightforward combination of existing techniques

2) Partly unsurprising results: it's already known that slot attention should work well for the RPM type of stimuli (e.g. discrete stimuli, low background clutter, limited occlusion, etc.)

3) [Minor]  Somewhat limited scope: object-centric approach implies it's ultimately limited to visual reasoning, as opposed to more generic abstract reasoning (e.g. numbers in RPM format).





**Note From Pc:**

if the above contains the word "oral" or "spotlight" please see: "oral" presentation means -> notable-top-5% and "spotlight" means -> notable-top-25%. As stated in our emails, we are disassociating presentation type from AC recommendations

**Summary Of Ac-Reviewer Meeting:**

The reviewers were unanimous in their lukewarm, borderline view of this paper. The above strengths and weaknesses were echoed by all the reviewers, and none had any strong feelings about whether the paper should be accepted or rejected.

This was a tough call. On one hand, very limited technical novelty ultimately made the paper unexciting to all, including myself. On the other hand, SOTA performance is an achievement in itself, and one could argue the conceptual simplicity of the paper is a positive rather than a negative.

Ultimately, my recommendation was based on the authors' framing of the paper. To their credit, they tackle the issue about technical novelty head-on. They clearly acknowledged it, but counter that the contribution is more a scientific one: the demonstration that these common, standard techniques put together can do as well as, or better than, other more sophisticated and involved techniques. To use an extreme analogy in my own words: if a paper showed that some form of simple nearest neighbors (i.e. zero technical novelty) could outperform transformers on a variety of standard datasets, then surely that is a result worth sharing, and a clear contribution to the community.

Additionally, I view this paper's contributions from a few different lenses:
1) A "scientific" paper in the sense of actually showing proof of something, even if that might supposedly seem unsurprising
2) A "workhorse" paper in the sense of contributing to the continuous pushing of SOTA performance for the field
3) A "strong baseline" paper in the sense of a simple, straightforward method that others should in the future should compare to as an example of something simple that works.